# Distinct spatially organized striatum-wide acetylcholine dynamics for the learning and extinction of Pavlovian associations

Safa Bouabid [1,2], Liangzhu Zhang [1], Mai-Anh T. Vu [1,2], Kylie Tang [1], Benjamin M. Graham[1], Christian A. Noggle[1] & Mark W. Howe [1,2] ✉

Striatal acetylcholine (ACh) signaling is thought to counteract reinforcement signals, promoting extinction and behavioral flexibility. Changes in striatal ACh signals have been reported during learning, but how ACh signals for learning and extinction are spatially organized to enable region-specific plasticity is unclear. We used array photometry in mice to reveal a topography of opposing changes in ACh release across distinct striatal regions. Reward prediction error encoding was localized to specific phases of ACh dynamics in anterior dorsal striatum (aDS): positive and negative prediction errors were expressed in dips and elevations respectively. Silencing ACh release in aDS impaired extinction, suggesting a role for ACh elevations in down-regulating cue-reward associations. Dopamine release in aDS dipped for cues during extinction, inverse to ACh, while glutamate input onto cholinergic interneurons was unchanged. These findings pinpoint where and suggest an intrastriatal mechanism for how ACh dynamics shape region-specific plasticity to gate learning and promote extinction.

Cholinergic interneurons (CINs) supply the main source of acetylcholine (ACh) to the striatum, the principal input nucleus of the basal ganglia[1,2]. ACh release is capable of modulating the excitability and plasticity of striatal spiny projection neurons to influence behavior on timescales ranging from 100 s of ms to days[3,4]. Electrophysiological studies have identified putative CINs by their characteristic tonically active firing profiles to investigate their natural dynamics in behaving rodents and primates during associative learning[5,6]. These studies revealed multi-phasic response profiles in CINs to primary rewards and reward predicting cues consisting of a short latency burst in firing, followed by a pause, then a post-inhibitory rebound[7–9]. Each of these components are variably expressed across CINs and can evolve in magnitude and duration as cues become associated with reward[8,10,11]. In particular, pauses in CIN firing to reward-predictive cues become larger and broader from the naive to learned stages of Pavlovian conditioning[8,11]. Larger cue-evoked CIN pauses for reward-associated cues are inverse in polarity to the coincident phasic elevations in cue-

evoked dopamine (DA) neuron firing and release, widely hypothesized to represent positive reward prediction errors (RPEs)[12–18]. Positive RPEs encoded in DA release are believed to promote striatal synaptic plasticity for associative learning, while coincident dips in ACh release are believed to set a permissive temporal window for DA driven plasticity to occur[19–21]. Evidence in slices and in-vivo suggest that DA release may itself promote larger ACh pauses through inhibitory D2 receptors on CINs[8,13,14]. However, studies using different associative learning paradigms have found little evidence that CIN (or ACh) pauses are fully consistent with RPEs for both predictive cues and rewards, indicating that DA elevations and ACh pauses are at least partially independent signals[10,15,22].

While in-vivo recordings of striatal ACh signaling have focused primarily on dynamics during initial associative learning, selective CIN manipulations in rodents have indicated roles in behavioral flexibility, response suppression, and extinction[23–33]. The specific effects of such manipulations vary across functionally distinct striatum regions. A key

---

¹Department of Psychological & Brain Sciences, Boston University, Boston, MA, USA. ²Aligning Science Across Parkinson's (ASAP) Collaborative Research Network, Chevy Chase, MD, USA. ✉e-mail: mwhowe@bu.edu

element of flexible behavior is the ability to down-regulate learned associations between previously valued cues and rewards when the associations are no longer valid, such as when a reward source is depleted. Dips in striatal DA release, putatively encoding negative prediction errors, arise to unexpectedly omitted rewards and to low predictive value cues and may promote the extinction of cue-reward associations[34–37]. However, although CINs have been implicated broadly in flexibility and extinction learning, few studies have attempted to examine the natural ACh dynamics as cues lose their reward predictive value.

One major challenge in understanding the dynamics of ACh signaling in-vivo is capturing variations in signaling across striatum regions. The striatum is a large, deep brain structure, and different functions in learning and action have been linked to particular striatal territories. Functional heterogeneity likely arises, at least in part, from topographically organized input from cortical and sub-cortical regions. Mounting evidence has indicated that anatomical hetero-geneity is also manifested in striatal ACh signaling during behavior. Although some studies have shown periods of high synchrony across CINs within a given region[6,8,13], others have reported significant signal differences across striatum regions[22,38–43]. Anatomical heterogeneities in ACh signaling have been difficult to assess due to the sparseness of CINs in the striatum (~3–5% of total population) and inherent limita-tions in spatial coverage of current optical and electrophysiological approaches. Standard electrode implants measure spiking from only a few unique CINs per subject, typically target limited striatal regions (<4), and without optical tagging can suffer from ambiguities in accurate CIN identification. Fiber photometry has been used to opti-cally measure ACh release, but similarly has been limited to 1–2 striatal regions per experiment. Therefore, it is poorly understood how and whether ACh release dynamics differ across the striatum volume to support distinct functions in associative learning and extinction.

To address these limitations, we applied a multi-fiber array tech-nique to optically measure rapid ACh release dynamics across over 50 locations simultaneously throughout the entire 3-dimensional stria-tum volume in head-fixed, behaving mice. Changes in multi-phasic ACh signals over learning and extinction were cue modality and signal component specific and varied across distinct anatomical gradients. Signaling at specific locations in the anterior dorsal striatum (aDS) was consistent with positive and negative RPEs for particular ACh signal phases and was opposite in polarity to classic DA RPEs. In support of a functional role for elevations in aDS ACh release in the down-regulation of cue-reward associations, silencing ACh release in the aDS impaired extinction but left initial learning intact. Finally, mea-surements of aDS DA and glutamate release onto CINs suggest an intrastriatal mechanism for the emergence of aDS ACh elevations. Overall, our results identify a spatiotemporal organization of striatal ACh signals positioned to gate region-specific plasticity during learn-ing and downregulate learned cue-reward associations during extinction.

## Results

### Striatum-wide measurements of ACh release with chronic micro-fiber arrays

We optically measured rapid ACh release dynamics across the striatum in head-fixed behaving mice using large-scale arrays of 55–99 optical micro-fibers[44] (50 μm diameter) distributed uniformly across the 3-dimensional striatum volume (Fig. 1a–e). Each fiber collected fluor-escence from post-synaptic neurons expressing the genetically enco-ded ACh sensor, GRAB-ACh3.0[45] (Fig. 1a). Based on previous empirical measurements and light scattering models[46,47], we estimate that our fibers collect fluorescence from a tapered collection volume extending approximately 100 μm axially and 25 μm radially from each fiber tip (see Vu et al., for details[44]). The minimum separation of fiber tips in our arrays was 220 μm radially and 250 μm axially ensuring no (or very

minimal) overlap in the collection volumes for each fiber. Fiber loca-tions in the brain were precisely reconstructed from post-mortem computerized tomography scans and aligned to a common coordinate framework to enable comparisons of spatial patterns of ACh release within and across subjects (Fig. 1d, e)[44,48]. Rapid, bi-directional changes in fluorescence, a proxy for changes in ACh release and synchronous firing of cholinergic interneurons[49], could be reliably detected across striatum locations (Fig. 1c). Unpredicted water reward deliveries pro-duced multi-phasic release profiles at some locations, consisting of an early peak, followed by a dip below baseline, then a later peak (Fig. 1f, g). However, all three signal components were not present in every region (Fig. 1g). The magnitudes of each component were gen-erally stable across days enabling us to account for potential changes in sensor expression or light collection during chronic recordings. Only a small fraction of fibers showed statistically significant changes in unpredicted reward mean ΔF/F across days of learning (1.6%, 5.7% and 5.7% of the 295 total fibers within the striatum for early peak, dip and late peak, respectively, $p < 0.01$, one-way ANOVA test comparing the mean of each day with the mean of every other day, see "Methods", Fig. 1f). Thus, our optical approach enabled us to investigate the evo-lution of simultaneous, spatiotemporal patterns of ACh release across the striatum during learning.

To control for potential signal contamination by hemodynamic or motion artifacts, we conducted measurements with quasi-simultaneous 405 nm illumination, the approximate isosbestic point of the sensor[45] and in mice expressing a non-functional null version of the ACh3.0 sensor (Fig. 1h, i). Rapid peaks and dips were generally not observed in these control conditions. However, some slow, low amplitude fluctuations were present, which could largely be elimi-nated with high pass filtering (see "Methods", Fig. 1i). In mice expres-sing the null sensor, we found that some small artifacts were present with 470 nm illumination which were not detected with 405 nm illu-mination, and overall, the artifactual fluorescence changes measured with 405 nm illumination were only weakly correlated with 470 nm illumination (Fig. 1h). Therefore, quasi-simultaneous 405 nm illumi-nation may not reliably account for all potential signal artifacts, likely because of differences in hemoglobin absorbance and tissue scattering between the two illumination wavelengths[50]. Thus, going forward, we used the null sensor expressing cohort with 470 nm illumination to rule out significant contributions of artifacts to behavior related sig-nals in ACh3.0 expressing mice.

### Spatially organized, opposing changes in striatal ACh release during Pavlovian learning

To determine how and where learning related changes in cue evoked ACh release dynamics occur, we mapped changes in ACh release across the striatum as mice learned a dual-cue delay Pavlovian con-ditioning task. Head-fixed mice ($n = 8$) on a spherical treadmill[51] were trained to associate light (470 nm LED, 7 mW) and tone (12 kHz, 80 dB) stimuli with water reward after a fixed 3 s delay (Fig. 2a). Cues were presented one at a time and pseudo randomly interleaved with 4–40 s intertrial intervals (ITIs) drawn from a uniform distribution. Non-contingent rewards were delivered occasionally (8 per session) in ITI periods. During Pavlovian learning, mice optimized their licking in two ways: they increased the fraction of time spent licking during the cue period ($p = 0.0078$, two-tailed Wilcoxon signed rank test, Fig. 2b–d), and they decreased non-contingent, 'spontaneous' licking during the ITI period ($p = 0.0273$, one-tailed Wilcoxon signed rank test, Fig. 2e). A lick index was computed, reflecting the relative proportion of antici-patory licking within the cue period relative to the ITI, to assess learning of the cue-reward association on each session (Fig. 2c).

Consistent with previous findings, we observed, for some fibers, multi-phasic, positive and negative changes in ACh release to condi-tioned cues after learning, consisting of a fast latency ('early') peak, a dip, and a second longer latency ('late') peak following the dip

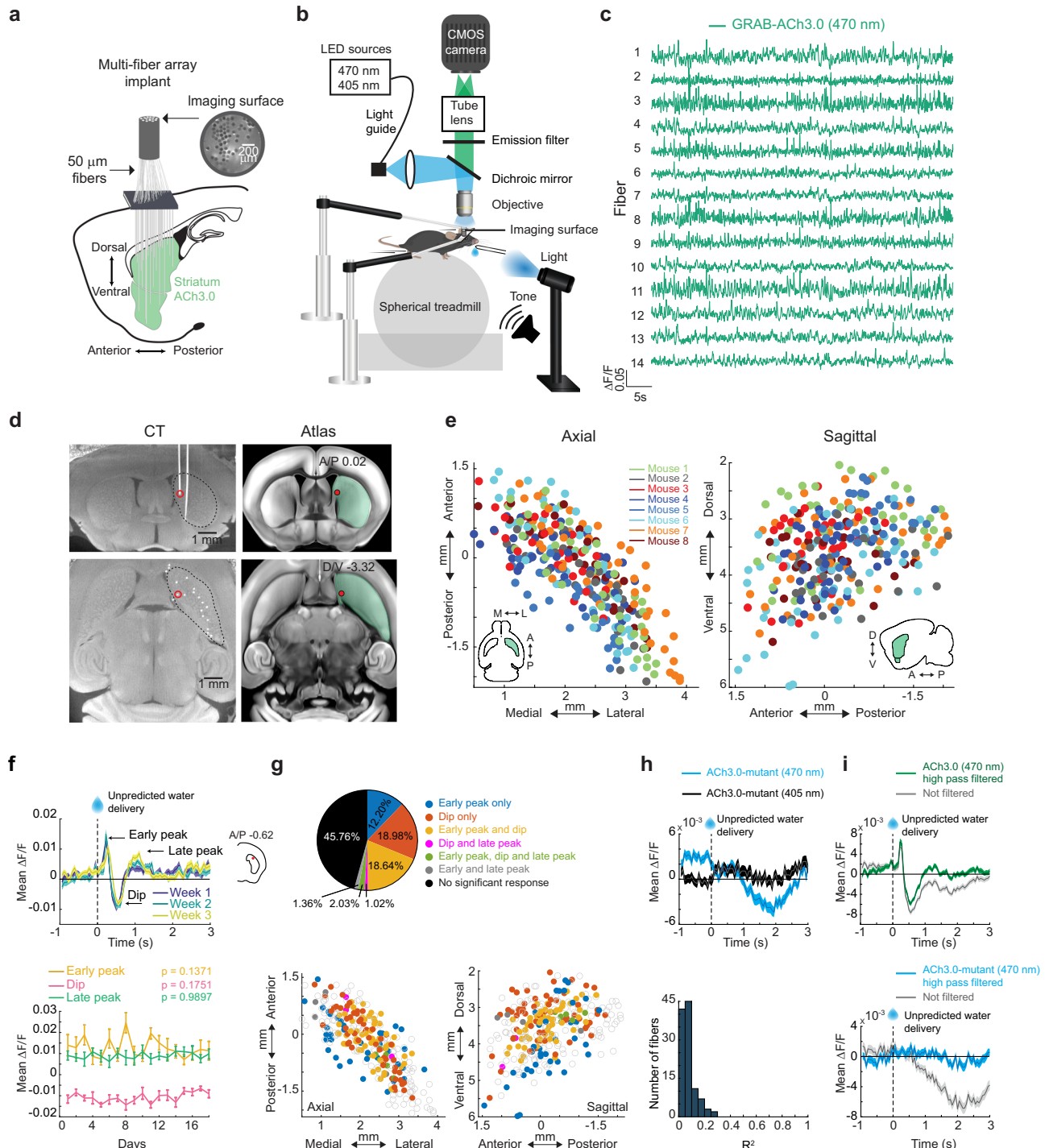

(Fig. 2f, h; Figure S1a, b). Prior to learning, cues evoked multi-phasic ACh release at many fewer fibers for all three components (Fig. 2f–i; Figure S1a–d). Early ACh peaks emerged and increased in magnitude with learning for a large fraction of fibers for both cues (light: 61.69%, tone: 51.53% of total fibers), while a smaller fraction exhibited peaks that remained stable across learning (Fig. 2f, g; Figures. S1c, d and S2c). In contrast, ACh dips for both cues predominantly became larger with learning (Fig. 2h, i; Figure S2c, light: 46.78%, tone: 39.66% of total fibers). Late ACh peaks also emerged, but for a much larger fraction of locations for the light cue than the tone (Figure S2a, c; light: 37.63%, tone: 8.47% of total fibers). These learning related changes were consistently present across mice expressing the ACh3.0 sensor but were not observed in mice expressing the non-functional mutant sensor,

ruling out significant contributions of hemodynamic or motion artifacts (Figure S2j).

To visualize anatomical patterns in the evolution of cue-evoked ACh release across the striatum, we constructed maps for each component reflecting spatially weighted averages of release changes during learning across fibers (see "Methods"). Signal changes for the early peak and dip were concentrated across two spatially segregated, partially overlapping regions (Fig. 2j, k; and Supplementary Fig. S2d). For both cues, the largest increases in the early peak were concentrated in a region of the ventral striatum (VS), near the center of the striatum in the A/P axis (Fig. 2j; and Supplementary Figs. S1e, g and S2d–f). While some early peak changes were present in the anterior dorsal striatum (aDS), the changes were smaller and peaks had shorter latencies

**Fig. 1 | Chronic striatum-wide measurements of rapid ACh release dynamics in head-fixed, behaving mice. a** Schematic of the fiber array approach for measuring ACh release across the striatum. **b** Spherical treadmill and imaging set up for head-fixed mice. **c** ACh3.0 fluorescence traces (ΔF/F) measured from 14 example fibers in the striatum of a single mouse. **d** Left: Post-mortem micro-CT scan images in the coronal plane (top) and axial plane (bottom) from a representative mouse. Fibers appear in white. Right: Images from the Allen Brain Common Coordinate Framework Atlas corresponding to the CT planes on left. Red circles indicate the position of an automatically localized fiber tip. **e** Locations of all fibers used for ACh measurements in the axial (left) and sagittal (right) planes. Each dot is the location (relative to bregma) of a single fiber and the color indicates the mouse identity (*n* = 8 mice). **f** Top: mean ΔF/F aligned to unpredicted water reward delivery for a single fiber from a single mouse for single sessions (*n* = 8 reward deliveries/session) at 1–3 weeks from the first day of Pavlovian training (Fig. 2). Inset indicates fiber location in the coronal plane. Shaded regions, S.E.M. Bottom: The mean peak (early/ late) or dip ΔF/F for unpredicted reward deliveries across 18 consecutive sessions of training (*n* = 8 reward deliveries/session) for the example at top. Values are the mean ± S.E.M. For each component, the mean of each day was compared to the mean of every other day using one-way ANOVA, $p = 0.1371$ and $F_{(17,129)} = 1.420$ for early peak, $p = 0.1751$ and $F_{(17,129)} = 1.346$ for dip, $p = 0.9897$ and $F_{(17,129)} = 0.367$ for late peak. **g** Top: percentage of the total fibers (295 fibers) across all mice (*n* = 8) with significant ($p < 0.01$, two-tailed Wilcoxon rank-sum test, see "Methods")

component(s) to unpredicted reward delivery. Bottom: maps as in (**e**) showing the presence of each combination of signal components to unpredicted reward delivery for each fiber. Empty circles indicate no significant response. **h** Top: Mean ΔF/F aligned to unpredicted reward delivery with quasi-simultaneous 405 nm (black) and 470 nm (blue) illumination in a mouse expressing the non-functional mutant ACh3.0 sensor. Shaded regions, S.E.M. Bottom: Histogram of Pearson's correlation coefficients between ΔF/F fluorescence traces obtained with quasi-simultaneous 405 nm and 470 nm illumination in mutant ACh sensor expressing mice (*n* = 3). **i** Mean unpredicted reward triggered ΔF/F from a single fiber in functional ACh3.0 sensor (top) and mutant ACh sensor (bottom) expressing mice before and after 0.3 Hz high pass filtering. Shaded region, S.E.M. Note that the small, slow artifactual decrease in the mutant sensor recording is largely eliminated with filtering, but the rapid reward triggered release measured with the functional sensor is preserved. Brain schematic in (**a**), mouse schematic in (**b**), objective in (**b**), and water drop schematics in (**f**), (**h**), and (**i**) were adapted from SciDraw (scidraw.io) and are licensed under CC BY 4.0 (https://creativecommons.org/licenses/by/4.0/). Source (https://doi.org/10.5281/zenodo.3925911, https://doi.org/10.5281/zenodo.3925913, https://doi.org/10.5281/zenodo.4914800, and https://doi.org/10.5281/zenodo.3925935), respectively. Brain schematics in (**e**) and (**f**) were adapted from the Allen Mouse Brain Common Coordinate Framework (CCFv3) (https://atlas.brain-map.org/). Source data are provided as a Source data file.

compared to the VS (Fig. 2j; and Supplementary Figures. S1e–g and S2d–f). In contrast to the early peak, the largest changes in the dip were concentrated in the aDS for the light but shifted slightly posterior for the tone (Fig.2k; and Supplementary Fig. S2d–f). Dip changes also extended partially into the most anterior portion of the VS (Supplementary Fig. S1g and Supplementary Fig. S2e,f). Increases in the late peak overlapped with dip changes for the light cue in the aDS, but the sparse late peaks for the tone cue were confined to a restricted region of the posterior striatum (Supplementary Fig. S2b, d–f). Variation in the component changes across striatal axes was quantified using a generalized linear model which included a mouse identity term to account for differences across individual mice (Fig. 2l, m, see Methods). Spatial gradients for early peaks and dips differed significantly across distinct striatal axes (Fig. 2l). In general, more significant components and larger changes with learning were present for the light cue relative to the tone (Supplementary Fig. S1a–d and S2c, f–h). Additionally, gradients for tone and light cues differed significantly along at least one axis for all three signal components, and cue selective changes were present for individual fibers in distinct territories perhaps reflecting the organization of excitatory input to the striatum[52,53] (Fig. 2m; Figure S2g). In summary, the acquisition of Pavlovian associations is accompanied by bi-directional (larger peaks, larger dips), sensory modality specific changes in cue-evoked ACh release across distinct 3-dimensional striatal territories.

Changes in cue-evoked ACh release with learning are partly consistent, in some regions, with bi-directional encoding of positive RPEs, as widely reported in DA neuron firing and release[54–56]. We next asked whether signal changes at reward delivery reflected positive RPE encoding. Like the cues, water delivery was associated with sequential peaks and dips in ACh release (Fig. 3a, b; and Supplementary Fig. S3a, b). However, unlike early ACh peaks at the cue which almost exclusively increased with learning at a majority of sites, peaks at reward changed for a minority and displayed a mixture of increases and decreases (Fig. 3b). In contrast, reward-evoked ACh dips were significantly smaller for a majority of fibers (57.29%, Fig. 3a, b). Consistent with putative RPE encoding, the dip changes at reward were opposite to the changes observed for the cues (Fig. 2). Late peaks predominantly increased (inconsistent with RPE encoding), but weakly (Fig. 3b). The changes in reward dips from pre to post learning were consistent with comparisons between unpredicted (random) and predicted (cue associated) rewards, and for rewards following low vs high probability cues during extinction (Supplementary Fig. S3c, d). Diminished reward dips were observed

over a broad region of the dorsal striatum extending across the regions showing cue-evoked dip changes for the light and tone cues (Fig. 3c; and Supplementary Fig. S3a, b). Increases and decreases to the peaks were scattered, with few concentrated in the VS regions showing strong elevations in the cue peak (Fig. 3c). Overall, only a relatively small proportion of fibers (light: 13.9%, tone: 9.8% of total fibers) had both increases in ACh early peaks to the cue and decreases in peaks to the reward (indicative of positive RPE encoding), with the majority increasing exclusively for cues (Fig. 3d, e; Figure S3e). For the dip component, a larger fraction of fibers (light: 29.2%, tone: 28.1% of total fibers) changed for both the cue and reward (smaller dip at reward, larger at cue) relative to cue or reward only, consistent with positive RPE encoding (Fig. 3d, e; and Supplementary Fig. S3e). Fibers with putative RPE encoding for the dip were localized to the aDS for the light cue and scattered across the more posterior dorsal striatum for the tone (Fig. 3d, e; and Supplementary Fig. S3e). Significant spatially organized changes were largely absent in null-sensor mice (Supplementary Fig. S3f). In summary, our results indicate that putative positive RPEs to cues and rewards are encoded predominantly by ACh dips within restricted, sensory modality specific regions of the dorsal striatum.

**Emergence of spatially organized ACh signals during extinction**
We next tested how and where cue-evoked ACh release changes as cues lose their predictive value during partial extinction learning. Following initial Pavlovian conditioning, the reward probability associated with one of the two cues was downshifted to 20% and the other to 80% (Fig. 4a). Cue-evoked anticipatory licking decreased over days specifically for the lower probability cue, indicating successful cue-specific extinction learning, at which point, reward probabilities for each cue were reversed (Fig. 4b–d). ACh early peaks predominantly decreased relative to the post learning phase for the low probability cue (light: 36.27%, tone: 40.34% of total fibers, Fig.4e, f; and Supplementary Fig. S4c) and dips became smaller (light: 22.03%, tone: 29.5% of total fibers, Fig. 4g, h; and Supplementary Fig. S4c). The direction of these changes represented a reversal of the changes during initial learning (Fig. 2, and Supplementary Fig. S2). However, unlike the early peaks and dips, ACh late peaks primarily increased for the low probability cue relative to late learning for both cues (light: 27.46%, tone: 20.68% of total fibers, Fig. 4g, h; and Supplementary Fig. S4c). Differences were consistent for the initial extinction and reversal phases, so data was combined for all analyses. These results indicate that the degradation of cue-reward associations is accompanied by a reversal

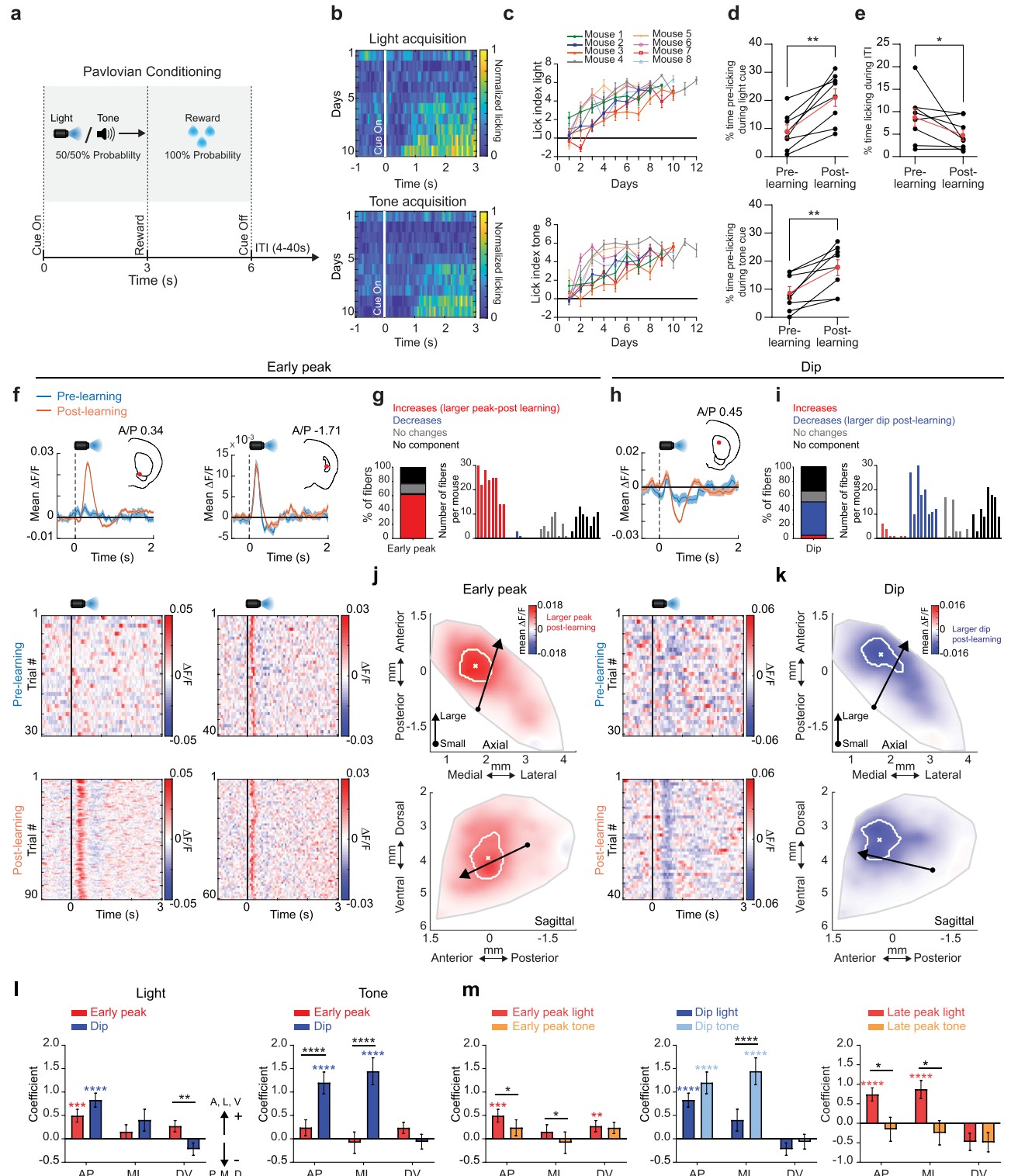

of learned dynamics for early peak and dip components but an emergence of new ACh increases for the late peak component.

We then examined the spatial organization of the changes in each component during extinction. The reversal of the early peak and dip changes were concentrated in largely the same regions of the VS and aDS (for the light cue) respectively as the regions showing changes during initial learning (Figs. 2j, k and 4i, j; and Supplementary Fig. S4a–c). The increases in the late peaks were concentrated almost exclusively in a restricted region of the aDS (Fig. 4j, and Supplementary Fig. S4a–c). These regional patterns held for differences in signaling

between the 20% and 80% probability phases for a given cue modality (Figure S5). The spatial gradient of the late peak changes was significantly different from that of the early peak changes for both cue modalities (Fig. 4k). Moreover, the spatial distribution of changes differed significantly between tone and light cues for the dip and late peak but not for the early peak (Fig. 4l; and Supplementary Fig. S4e,f). The region-specific emergence of late peaks in aDS occurred many trials before the decreases in anticipatory spout licking to the 20% probability cue (light cue: median= 29 trials before licking decrease, STD = 27, Fig. 4m, n; tone cue: median = 2 trials before licking decrease,

**Fig. 2 | ACh release components for conditioned cues evolve bi-directionally over distinct anatomical gradients during Pavlovian learning. a** Schematic of dual-cue delay Pavlovian conditioning task. **b** Normalized mean lick spout contacts aligned to light (top) and tone (bottom) cue onset for 10 consecutive sessions from the start of training for one representative mouse. **c** Mean lick index on light (top) and tone (bottom) cue trials ($n = 20$ trials for each cue) for each day after the start of Pavlovian training for each mouse ($n = 8$). Error bars, S.E.M. **d** Mean percent time spent licking during the 3 s light (top) and tone (bottom) cue interval prior to reward for each mouse for sessions pre and post learning. Red points are means (± S.E.M.) across all mice ($n = 8$). For both light and tone, $W_8 = 36$, **$p = 0.0078$, two-tailed Wilcoxon matched-pairs signed rank test. **e** Same as (**d**) for licking during ITI periods ($W_8 = -28$, *$p = 0.0273$, one-tailed Wilcoxon matched-pairs signed rank test). **f** Top: Mean ΔF/F aligned to the light cue onset for 2 representative fibers for trials pre (blue) and post (orange) learning. Left example, significant early peak change; right, no change. Shaded regions, S.E.M. Red dots in insets indicate the fiber locations in the coronal plane. Bottom: Light-cue-aligned ΔF/F for all trials included in the triggered averages at top. **g** Left: percent of all fibers with significant increases or decreases, no change, or no significant component from pre to post learning for the early peak ΔF/F at light cue onset. Right: histogram of number of fibers per mouse with significant early peak changes. Each bar is the fiber count for one mouse for each condition indicated by colors at left. **h** Mean ΔF/F aligned to the light cue onset as in (**f**) for a representative fiber with a significant change in the dip with learning. **i** Same as (**g**) for dip ΔF/F changes from pre to post learning. **j** Maps

(axial, top; sagittal, bottom) showing spatially weighted means across locations of differences with learning (post-pre ΔF/F) for the mean early peak ΔF/F at light cue onset. Lines indicate the axes of maximal variation and arrows indicate the direction of peak increases from smallest to largest changes. White contours indicate regions with changes in the highest 10th percentile. **k** Same as (**j**) for the dip component. **l** Model coefficients indicating the relative magnitude and direction of the variation in mean ΔF/F differences with learning for the peak and dip components for light and tone cues across each striatal axis (AP: anterior-posterior, ML: medial-lateral, DV: dorsal-ventral). The coefficients represent fixed effects derived from a linear mixed-effects model, with individual recording sites as the unit of analysis and mouse identity included as a random effect. The sign of the coefficient indicates the direction of the largest differences (see arrows). Error bars, S.E.M. *$p < 0.05$, **$p < 0.01$ ***$p < 0.001$, ****$p < 0.0001$, two-tailed Wald t-test on model coefficients followed by bonferroni *post hoc* analysis on model coefficients. Significant interaction terms (black) indicate difference in coefficients between peak and dip for a given axis. **m** Same as (**l**) but comparing spatial coefficients for tone and light for all three components. The exact *p*-values and additional statistical details can be found in Supplementary Table 1. Water drop schematic in (**a**) adapted from SciDraw (scidraw.io), licensed under CC BY 4.0 (https://creativecommons.org/licenses/by/4.0/). Source (https://doi.org/10.5281/zenodo.3925935). Brain schematics in (**f**) and (**h**) were adapted from the Allen Mouse Brain Common Coordinate Framework (CCFv3) (https://atlas.brain-map.org/). Source data are provided as a Source data file.

STD = 15, Fig. 4o, p), indicating that these signals are positioned to drive learning related plasticity, perhaps contributing to behavioral extinction (see Discussion). Again, significant changes during extinction were not present in null sensor expressing mice (Supplementary Fig. S4d). Overall, these findings indicate that flexible downshifting of learned cue values is accompanied by distinct, bi-directional changes in cue-evoked ACh release across different striatal regions that precede a downregulation of behavior. Most notably, aDS late peaks did not revert back to a pre-learning state (as for peaks in the VS) but acquired new elevations reflecting the relative (negative) change in predictive cue value.

Increases in ACh late peaks for the low value cues during extinction are consistent with the emergence of negative RPE encoding reflecting either the negative change in predictive cue value from the learned phase or the relative difference between the high and low value cues. The aDS region exhibiting the emergence of cue-evoked late peaks during extinction aligned with the region encoding positive RPEs in ACh dips to light cues and rewards during initial learning (Fig. 3d), suggesting that the aDS may preferentially encode both positive and negative RPEs through ACh dips and late peaks respectively (see Discussion). Our experimental design did not allow us to unambiguously resolve putative negative RPE signals for reward omissions during the extinction phase, since mice were not required to precisely estimate the timing of reward delivery, and anticipatory licking began prior to the time of expected reward. However, alignment to reward omission could be tested during ITI periods, when mice spontaneously initiated spout licks when water was not present. Mice initiated spontaneous spout licking bouts in the ITI less frequently after learning the cue-reward association (Fig. 2e). This indicated that they predict a higher probability of random reward in the ITI early prior to learning the task structure. Therefore, unrewarded, spontaneous spout licks in the ITI would result in a larger negative RPE following the lick onset early in learning. Lick bout initiations in the ITI pre learning were associated with multi-phasic changes in ACh release at distinct latencies (Figure S6a,b). Short latency increases (early peaks) in ACh began before the tongue contacted the lick spout, around lick initiation (Fig.5a). Dips were present at intermediate latencies, while longer latency increases (late peaks) occurred several hundred milliseconds after spout contact (Fig. 5c). All three components changed with learning for a significant proportion of fibers. Early peaks and dips predominantly decreased (Fig. 5b, early peak: 40%, dip: 26.1% of total fibers), while late peaks showed a mixture of increases

and decreases across fibers (Figs. 5d, 23.1% and 18% respectively). Early and late peaks were concentrated across largely distinct striatum locations prior to learning. Early peaks were prominent in the posterior striatum, while late peaks were localized primarily to the aDS (Supplementary Fig. S6b) and peaks in both regions decreased with learning (Fig. 5e, f; and Supplementary Fig. S6c). Some late peaks also emerged and became larger with learning, but in a distinct region of the central lateral striatum (Fig. 5f; and Supplementary Fig. S6c). Dips became larger with learning across the dorsal striatum (Fig. 5e, and Supplementary Fig. S6c). The changes in the ACh late peaks aligned in the aDS with the late peak elevations to the downshifted cues during extinction (Fig. 5g). Together, these results indicate that ACh release in the aDS signals putative negative RPEs through elevations in long-latency ACh peaks and positive RPEs through dips (Fig. 5g, h).

## Changes in ACh release during learning and extinction cannot be explained by locomotion changes

ACh signaling has been previously linked to changes in locomotion[39,49,57,58], so we asked whether changes we observed across learning phases could be explained by changes in treadmill velocity. A generalized linear model (GLM) was used to partially account for potential correlations with continuous linear velocity and acceleration (see Methods), but we conducted additional analyses to address potential movement contributions not captured by the GLM. During initial Pavlovian learning, mice decelerated significantly more, on average, after cue onset and less after reward delivery as they learned the cue-reward associations (Supplementary Fig. S7a, b, $p > 0.05$ for peak deceleration and $p < 0.05$ for velocity changes after cue onset, Freidman test; $p < 0.05$ for peak deceleration and $p > 0.05$ for velocity after reward delivery, Wilcoxon rank-sum test). Differences were not present over learning for spontaneous ITI lick bouts (Supplementary Fig. S7c, $p > 0.05$, Wilcoxon rank-sum test) or during extinction for high and low probability cues or relative to the late learning phase (Supplementary Fig. S7a, $p > 0.05$, Freidman test). To address the possibility that locomotion deceleration signals may contribute to the region-specific ACh changes that we report with initial learning (Figs. 2 and 3; and Supplementary Fig. S1–3), we first examined ACh release to spontaneous decelerations in the ITI period with similar magnitude to decelerations at cues and rewards (Supplementary Fig. S7d–g). At some locations, primarily in the dorsal striatum, we observed bi-phasic peaks and dips in ACh release associated with ITI decelerations (Supplementary Fig. S7d). However, these peaks and

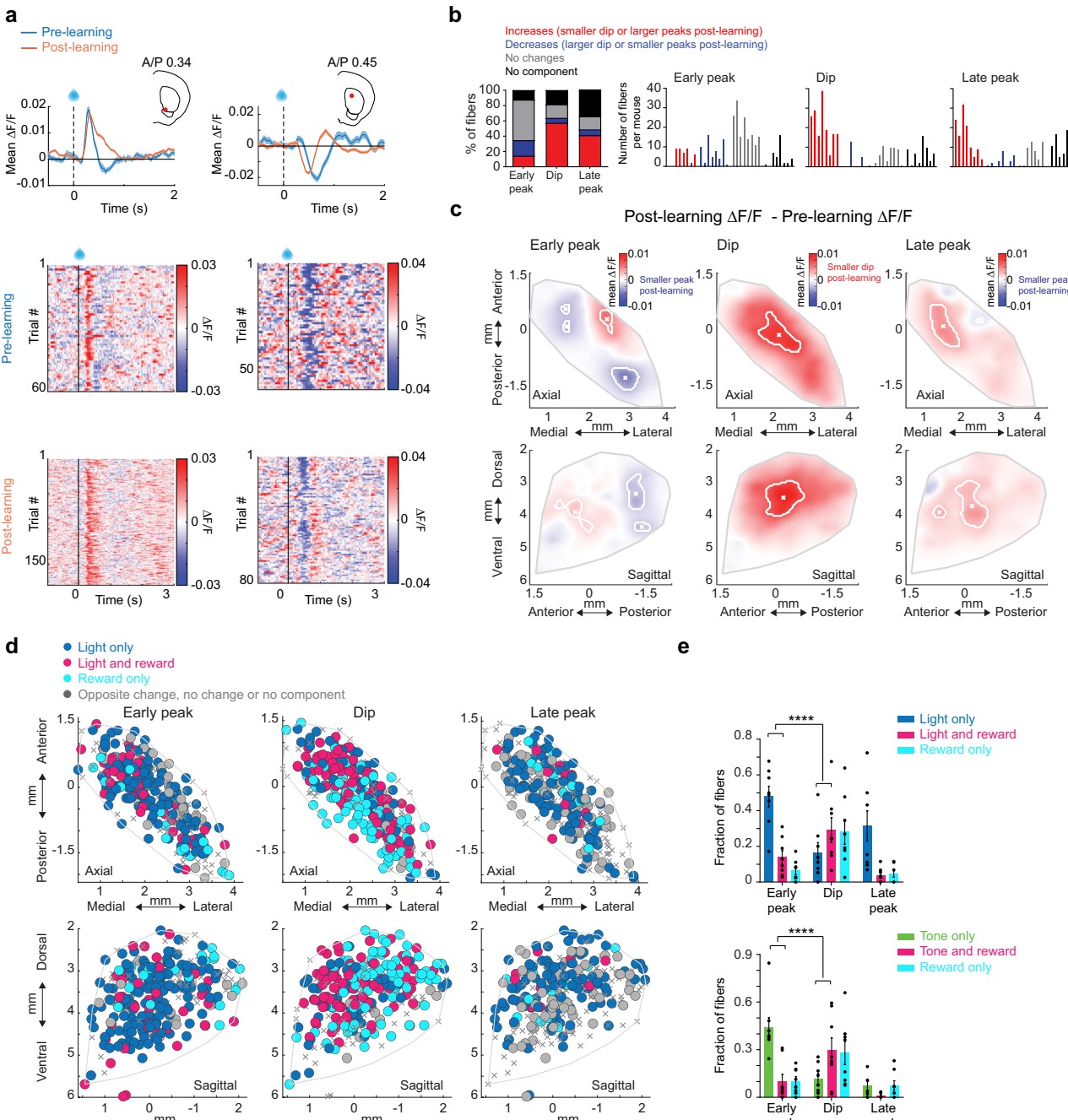

**Fig. 3 | Opposite ACh changes with learning at cue and reward are signal component selective and spatially concentrated in the anterior dorsal striatum. a** Top: Mean ΔF/F aligned to the reward delivery in the task for 2 representative fibers (same fibers as Fig. 2f, left and 2 h) for trials pre (blue) and post (orange) learning. Significant changes for both are present in the dip, but not the early peak. Shaded regions, S.E.M. Red dots in insets indicate the fiber locations in the coronal plane. Bottom: Reward-triggered ΔF/F for all trials included in the triggered averages at top. **b** Left: percent of all fibers with significant increases or decreases, no change, or no significant component from pre to post learning for each signal component ΔF/F at reward consumption onset. Right: histograms of the number of fibers per mouse with significant changes for each component. Each bar is the fiber count for one mouse for each condition indicated by colors at left. **c** Maps (axial, top; sagittal, bottom) showing spatially weighted means across locations of differences with learning (post-pre) for the mean ΔF/F for the three signal components at reward consumption onset. White contours indicate regions with changes in the highest 10th percentile. **d** Maps showing each fiber (dot) color coded according to whether significant changes from pre to post learning (see "Methods") were present at the light cue onset only (dark blue), reward consumption only (light blue) or both (pink). Pink dots indicate locations where the component magnitude became larger (dip more negative or peak more positive) for cue and smaller for reward over learning, consistent with reward prediction error encoding. **e** Fraction of all fibers for each component classified according to changes across learning for light cue (top) or tone cue (bottom) and reward as indicated in (**d**). Data are presented as the mean ± SEM, with each dot representing one mouse (n = 8). (****p = 8.1 × 10⁻¹³ early peak vs. dip between light only and light and reward, ****p = 7.8×10⁻¹⁸ early peak vs dip between tone only and tone and reward, ****p < 0.0001, Fisher's exact test, two-sided). Water drop schematic in (**a**) adapted from SciDraw (scidraw.io), licensed under CC BY 4.0 (https://creativecommons.org/licenses/by/4.0/). Source (https://doi.org/10.5281/zenodo.3925935). Brain schematic in (**a**) was adapted from the Allen Mouse Brain Common Coordinate Framework (CCFv3) (https://atlas.brain-map.org/). Source data are provided as a Source data file.

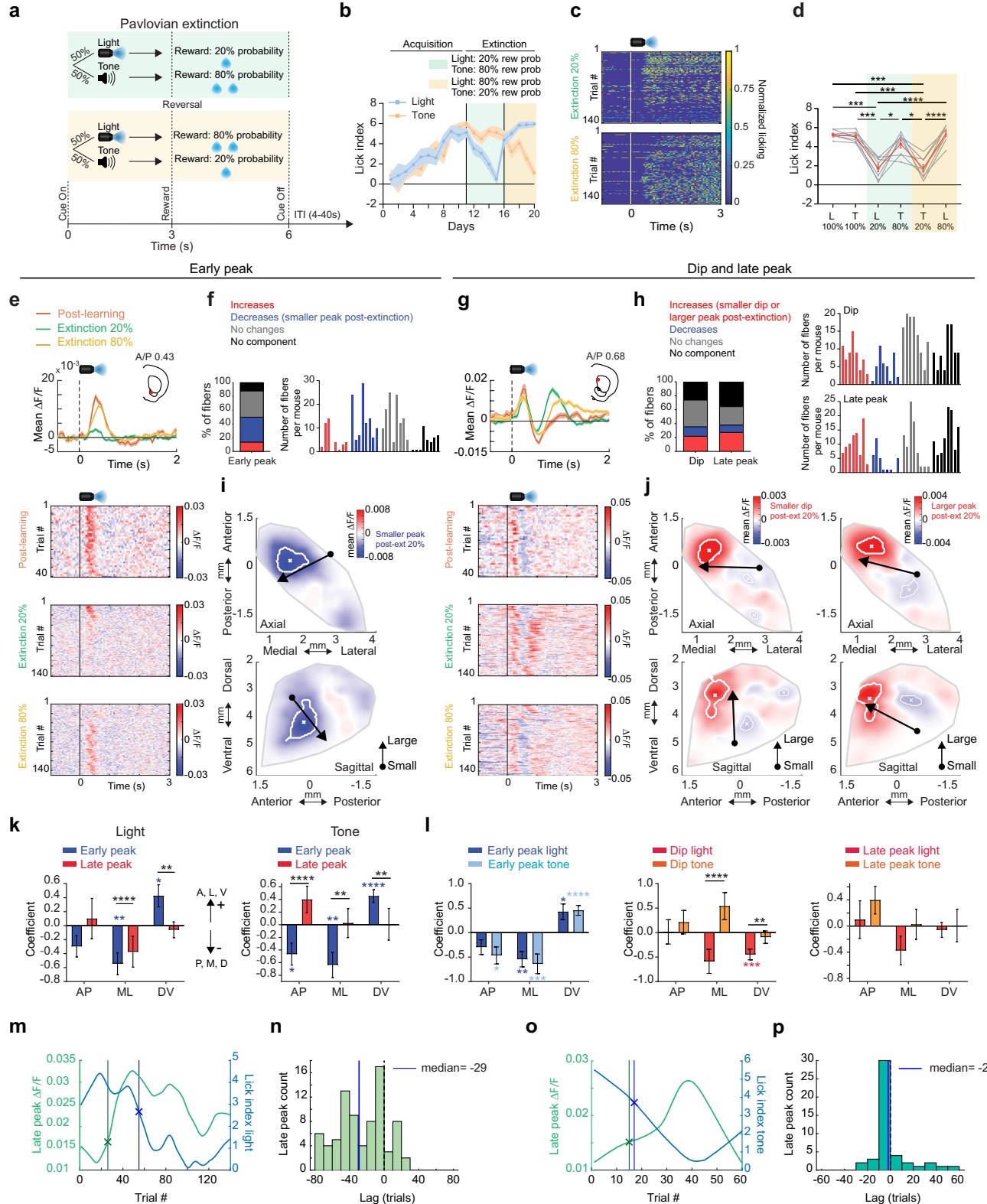

dips in ΔF/F were present across fewer fibers and were much lower in magnitude relative to changes to cues and rewards with learning (Figure S7e–g). Most fibers with large cue and reward changes with learning did not have any significant ITI deceleration response (36.15% and 23.76% of fibers with significant peak and dip changes at cue had significant maxima and minima for ITI decelerations respectively; 50% and 25.43% for fibers with peaks and dips at reward respectively; Supplementary Fig. S7f, g). In addition, change magnitudes at cue and

reward with learning were not correlated with magnitudes of peaks and dips to spontaneous decelerations across fibers (Supplementary Fig. S7l, m; and Supplementary Table 2). For most fibers, early peaks and dips for cues and rewards were highly consistent in timing and amplitude across trials within a given session, despite significant variability in the timing and magnitude decelerations at cue and reward (Figs. 2, 3; and Supplementary Fig. S7h, i). Moreover, ACh release to cues, rewards, and ITI licks were mostly insensitive to trial-

**Fig. 4 | Emergence of cue-evoked ACh increases in the anterior dorsal striatum during partial extinction. a** Schematic of the partial extinction paradigm. **b** Mean lick index on light and tone cue trials for each day of training for one representative mouse. Shaded regions, S.E.M. across trials for each day. **c** Rasters of normalized licking aligned to light cue onset for all trials from the beginning of extinction training for a single mouse. **d** Lick indices across all mice ($n = 8$) for the light and tone cues associated with different reward probabilities during initial learning (100%) and partial extinction (20% and 80%). Mean across mice shown in red, individual mice in gray. Error bars, S.E.M. Repeated measures one-way ANOVA, followed by Tukey *post hoc* test. *: $p < 0.05$, ***: $p < 0.001$, ****: $p < 0.0001$. T: tone, L: light. **e** Top: Mean ΔF/F aligned to the light cue onset for a representative fiber with a significant decrease in the early peak from learning to extinction for trials where light was associated with 100% (post-learning), 80%, and 20% (extinction) probabilities. Shaded regions, S.E.M. Red dots in insets indicate the fiber locations in the coronal plane. Bottom: Light-triggered ΔF/F for all trials included in the triggered averages at top. **f** Left: percent of all fibers with significant increases or decreases, no change, or no significant component from post learning to the 20% light extinction phase for the early peak ΔF/F at light cue onset. Right: histogram of number of fibers per mouse with significant early peak changes. Each bar is the fiber count for one mouse for each condition indicated by colors at left. **g** Mean ΔF/F aligned to the light cue onset as in (**e**) for a representative fiber with a significant increase in the late peak and decrease in the dip from late learning to the extinction phases. **h** Left: percent of all fibers with changes from post learning to extinction as in (**f**) for the dip and late peak components. Right: histogram of number of fibers per mouse with significant dip (top) and late peak (bottom) changes. Each bar is the fiber count for one mouse for each condition indicated by colors at left. **i** Maps (axial, top; sagittal, bottom) showing spatially weighted means across locations of differences between the 20% extinction and 100% post learning phases (20%−100% ΔF/F) for the mean early peak ΔF/F at light cue onset. Lines indicate the axes of maximal variation and arrows indicate the direction of peak decreases from smallest to largest changes. White contours indicate regions with changes in the highest 10th percentile. **j** Same is (**i**) but for dip (left) and late peak (right). **k** Model coefficients indicating the relative magnitude and direction variation of the mean ΔF/F differences between the 20% extinction and 100% post learning phases for the early and late peak components for light and tone cues across each striatal axis (AP: anterior-posterior, ML: medial-lateral, DV: dorsal-ventral). The coefficients represent fixed effects derived from a linear mixed-effects model, with individual recording sites as the unit of analysis and mouse identity included as a random effect. The sign of the coefficient indicates the direction of the largest differences (see arrows). Error bars, S.E.M. *$p < 0.05$, **$p < 0.01$ ***$p < 0.001$, ****$p < 0.0001$, two-tailed Wald t-test followed by bonferroni *post hoc* analysis on model coefficients. Significant interaction terms (black) indicate difference in coefficients between early and late peak for a given axis. **l** Same as (**k**) but comparing spatial coefficients for tone and light for all three components. **m** Mean light cue-evoked late peak ΔF/F (green) and lick index (blue) for all trials following the transition from 100% reward probability to 20% reward probability for a single fiber. The lines indicate where each measure significantly (CUSUM algorithm, see Methods) changed relative to the 100% probability phase. **n** Histogram showing the # fibers with relative latencies between the significant increase in light cue-evoked late peak ΔF/F and the decrease in lick index following high to low reward probability transitions. Vertical line indicates the median of the distribution. **o** Same as (**m**) but for tone. **p** Same as (**n**), but for tone. The exact *p*-values and additional statistical details can be found in Supplementary Table 1. Water drop schematic in (a) adapted from SciDraw (scidraw.io), licensed under CC BY 4.0 (https://creativecommons.org/licenses/by/4.0/). Source (https://doi.org/10.5281/zenodo.3925935). Brain schematics in (**e**) and (**g**) were adapted from the Allen Mouse Brain Common Coordinate Framework (CCFv3) (https://atlas.brain-map.org/). Source data are provided as a Source data file.

by-trial variations in the size of the associated deceleration (Supplementary Fig. S7h–k). ACh peaks and dips for only a small fraction of fibers with significant response components were significantly correlated with trial by trial deceleration for cues (7.4% of 270, 7.1% of /224, and 1.1% of /180 fibers for early peak, dip, and late peak of light cue respectively; 8.3% of 262, 5.1% of 176, and 3.9% of 76 locations for early peak, dip, and late peak of tone cue respectively; Pearson's correlation, $p < 0.01$) and rewards (19.1% of 287, 23.5% of 276, and 6.1% of 229 locations for early peak, dip, and late peak respectively; Pearson's correlation, $p < 0.01$). Thus, although changes in locomotion modulate ACh release at some striatal locations, locomotion related signaling per se could not account for the patterns of cue and reward evoked release that we observed.

### Intact ACh release in the aDMS is required for extinction of Pavlovian associations

Our ACh release measurements identified transient increases in cue-evoked ACh release in the aDS (late peaks), consistent with negative RPEs, which may contribute to downshifting the predictive value of cues and the expression of appetitive behaviors, such as licking, when reward probabilities change (Figs. 4 and 5). We tested whether suppressing ACh release altered spontaneous and cue evoked spout licking during Pavlovian learning and partial extinction by virally expressing Tetanus toxin light chain (TelC) selectively in cholinergic interneurons in the aDS of ChAT-cre mice. Targeting was informed by the distribution of late peak changes for the light cue (Fig. 4j), so injections were shifted slightly to favor the dorsal medial striatum (aDMS, see "Methods", Fig. 6a–c). TelC expression eliminates synaptic vesicle exocytosis in cholinergic interneurons, and consequently blocks ACh release[59] (Fig. 6c). ACh suppression did not affect the mice ability to learn the cue-reward association, as both TelC and control mice increased the frequency of licking in the cue period over learning (Two-way repeated measure ANOVA, $p = 0.02$ for Telc and $p = 0.0001$ for control, Fig. 6d). This is expected, given that the primary changes in ACh release in aDS during initial learning were in the dips, which may be permissive for learning[19,60–62] (Fig. 2i, k; and Supplementary Fig. S2c, d). During cue specific extinction, control mice exhibited a down-

regulation of anticipatory licking relative to the post learning phase, as expressed by the significant decrease in the fraction of time spent licking during cue period ($p = 0.04$, Two-way repeated measures ANOVA, Fig. 6d, f). However, TelC expressing mice did not show a significant down-regulation of cue period licking relative to the learned phase ($p = 0.93$, Two-way repeated measures ANOVA, Fig. 6d, f). TelC mice also had slightly higher licking rates during the ITI and cue periods than the control mice at all phases of learning, perhaps indicating a general difference in suppression of spontaneous licking (Fig. 6d, e). Although the manipulation was not temporally specific, these data indicate that intact ACh release in the aDMS contributes to the extinction of learned Pavlovian responses following a downshift in cue associated reward probabilities.

### Cue and unrewarded lick evoked dopamine dips opposite to ACh late peaks in aDS

We next investigated potential contributors to the emergence of the downshifted cue and spontaneous lick related ACh increases in the aDS. Dips in DA neuron firing and release have been measured in response to low value cues and unpredicted reward omissions (i.e. negative prediction errors) and have been proposed to contribute to extinction learning[34,35,54,63]. If present, these DA dips may disinhibit CINs through reduced tonic activation of inhibitory D2 receptors[13,59,64,65]. We tested whether DA signals in the aDS are consistent with a potential role in facilitating the observed elevations in ACh signaling. We optically measured DA release dynamics with dLight1.3b[66] in the same regions of the aDS where we observed the increases in ACh peaks to unrewarded licks and down-shifted cues (Figs. 4, 5, 7a). Consistent with prior work, DA release rapidly increased to conditioned cues following learning (Fig. 7b, c). Following partial extinction (Fig. 4a), dips below baseline emerged after short latency peaks, selectively for the low probability cue (Fig. 7b–d). The average latency to the trough of DA dips (median to trough = 0.63 s) preceded the average latency of the ACh late peaks (median = 0.8 s) recorded in the same aDS region in the ACh3.0 expressing group (Fig. 7e, $p = 0.02$, Two-tailed rank-sum test). Like the ACh late peaks, dips in cue-evoked aDS DA emerged within only a few trials after the probability shift, well

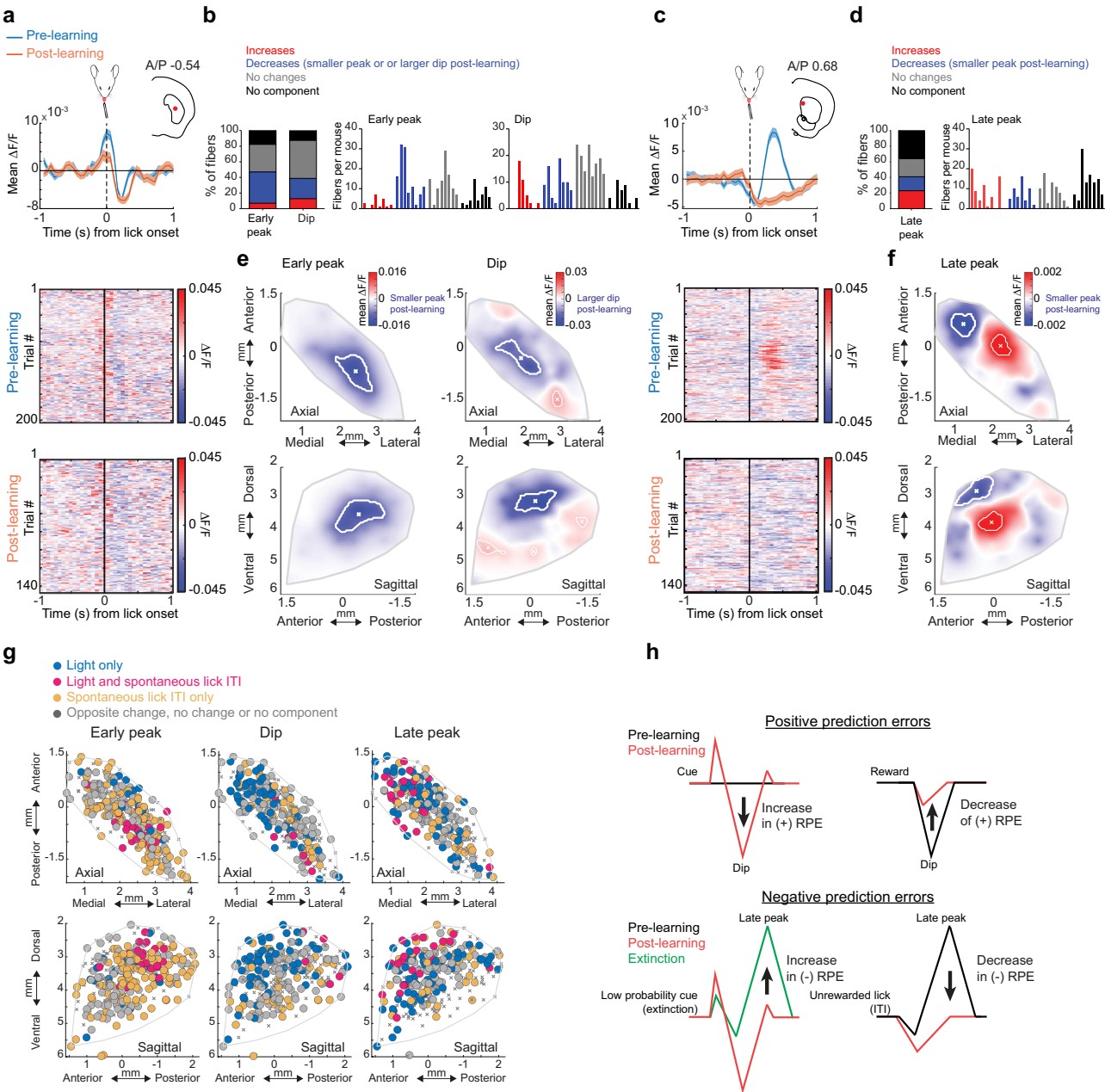

**Fig. 5 | Regional changes in ACh release components to unrewarded ITI licks during learning. a** Top: Mean ΔF/F aligned to unrewarded ITI lick bout onsets for one representative fiber with significant early peak and dip changes for trials pre (blue) and post (orange) Pavlovian learning. Shaded regions, S.E.M. Red dot in inset indicates the fiber location in the coronal plane. Bottom: Lick onset-aligned ΔF/F for all trials included in the triggered averages at top. **b** Left: percent of all fibers with significant increases or decreases, no change, or no significant component from pre to post learning for the early peak and dip ΔF/F at ITI lick bout onsets. Right: histogram of number of fibers per mouse with significant early peak and dip changes. Each bar is the fiber count for one mouse for each condition indicated by colors at left. **c** Same as (**a**) for a fiber with a significant late peak change. **d** Same as (**b**) for late peak changes with learning. **e** Maps (axial, top; sagittal, bottom) showing spatially weighted means across locations (see "Methods") of differences with learning (post-pre ΔF/F) for the mean early peak (left) and dip (right) ΔF/F at ITI lick bout onsets. White contours indicate regions with changes in the highest 10$^{th}$

percentile. **f** Same as (**e**) for late peak. **g** Maps (axial, top; sagittal, bottom) showing each fiber (dot) color coded according to whether significant changes from pre to post learning (see "Methods") were present at the light cue onset only (dark blue), unrewarded ITI lick only (orange) or both (pink). Pink dots indicate locations where the component magnitude was larger with extinction (dip more negative or peak more positive) for cue and smaller post learning for the unrewarded ITI lick, consistent with negative reward prediction error encoding. **h** Schematic summarizing the changes in ACh release in the aDS region with learning and extinction consistent with positive and negative RPE encoding for distinct signal components. Mouse head schematic in (**a**) and (**c**) adapted from SciDraw (scidraw.io), licensed under CC BY 4.0 (https://creativecommons.org/licenses/by/4.0/). Source https://doi.org/10.5281/zenodo.3925903. Brain schematics in (**a**) and (**c**) were adapted from the Allen Mouse Brain Common Coordinate Framework (CCFv3) https://atlas.brain-map.org/.) Source data are provided as a Source data file.

before the change in cue-evoked licking (median = 27 trials prior to lick change, STD = 23.4 for light, median = 22 trials, STD = 42.8 for tone, Fig. 7f). Like the ACh3.0 expressing mice, there were no differences in the dLight1.3b group in cue-related decelerations between the post

learning and extinction phases, so DA dips could not be explained by locomotion changes (Fig. 7g). We then asked whether DA release for spontaneous ITI licks also exhibited learning related dips reflecting shifts in negative RPE signaling with learning. DA release in the aDS

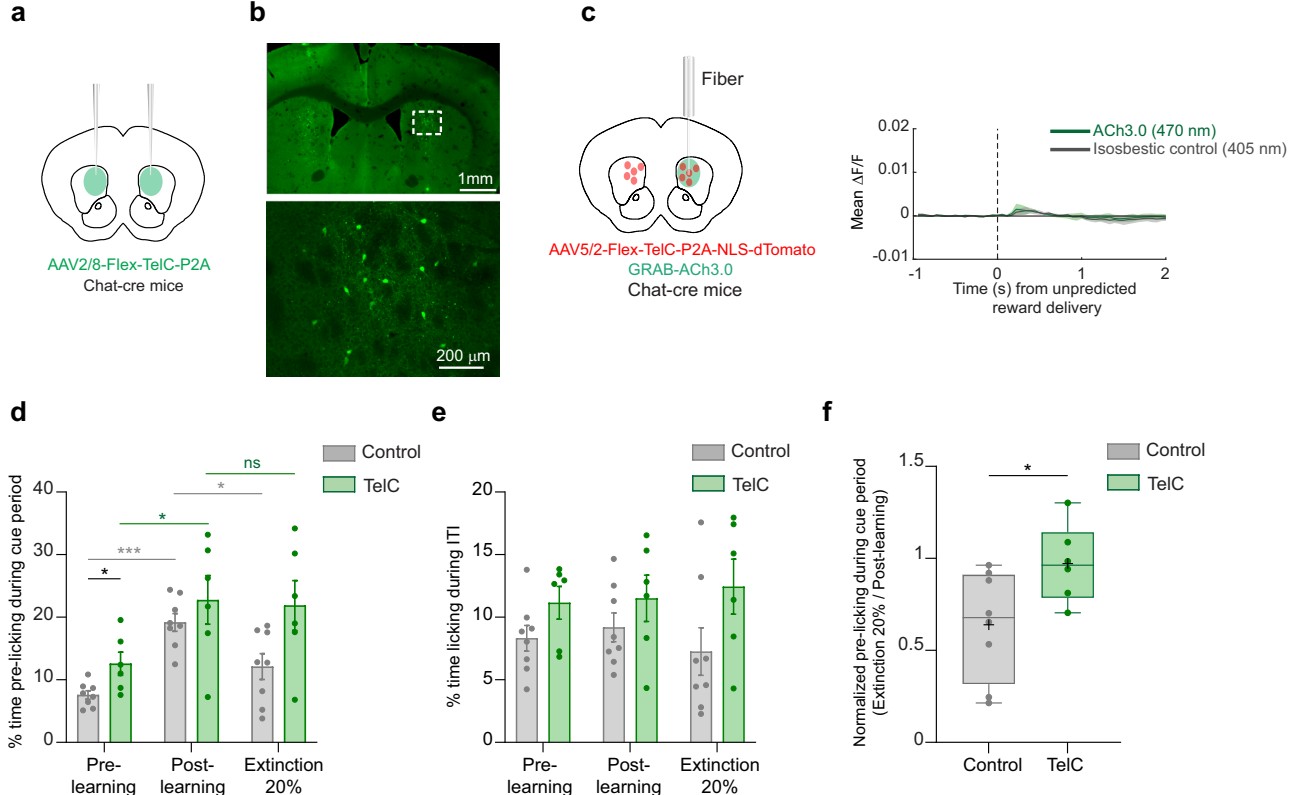

**Fig. 6 | Inactivation of ACh release in the anterior dorsal striatum impairs behavioral changes during extinction. a** Schematic showing the bilateral injection strategy for the selective expression of TelC in CINs in aDS. **b** Top: Fluorescence image in a coronal section of the striatum showing bilateral TelC expression in aDS for a representative mouse. Bottom: Magnification of the boxed area in top image showing selective expression of TelC in CINs. Experiments were independently repeated n = 6 times with similar results. **c** Left: Schematic showing the strategy for verifying effects of TelC mediated suppression of CINs on ACh release to unpredicted rewards. Right: Mean ΔF/F across all trials (n = 209) aligned to unpredicted reward delivery with quasi-simultaneous 405 nm (gray) and 470 nm (green) illumination in two mice expressing the ACh3.0 sensor and TeLC in CINs. Shaded regions, S.E.M. **d** Mean (± S.E.M.) % time spent licking during the 3 s of cue presentation (both light and tone cues merged) for pre learning, post learning and extinction (20% reward probability cue only) for both TelC and control mice. Each dot is an individual mouse (n = 8 for control and n = 6 for TelC mice). *$p < 0.05$,

***$p < 0.001$, two-tailed repeated measures two-way ANOVA followed by Tukey *post hoc* analysis for comparisons within and between TelC and control mice. **e** same as (**c**) for % time licking during ITI periods, data are presented as mean (± S.E.M.). The exact *p*-values and additional statistical details can be found in Supplementary Table 1. **f** Box plots showing the ratio of mean cue period licking during partial extinction (20% reward probability, light and tone combined) relative to post learning (100% reward probability), indicating no change in cue licking in TelC mice and a reduction in normalized cue licking in control mice during partial extinction. Each dot represents an individual mouse (n = 8 for control and n = 6 for TelC mice). Box plots with whiskers extending from the minimum to the maximum values. The center line represents the median, the box bounds show the 25th and 75th percentiles, and the + symbol indicates the mean. Mann-Whitney U test, U$_{(8,6)}$ = 7, *$p = 0.0293$ (two-tailed). Brain schematics in (**a**) and (**c**) were adapted from the Allen Mouse Brain Common Coordinate Framework (CCFv3) (https://atlas.brain-map.org/). Source data are provided as a Source data file.

---

increased for spontaneous licks, rising just prior to spout contact (Fig. 7h, i). Early in Pavlovian learning, peaks were followed by dips below baseline, but these dips largely disappeared after learning ($p = 2.6 \times 10^{-3}$, two-tailed Wilcoxon rank-sum test), mirroring opposite changes in ACh release as ITI licks became less frequent (Fig. 7h–j). Like the cue responses, DA dips had a shorter average latency (median = 0.33 s) relative to the ACh late peaks (median = 0.44 s, $p = 3.33 \times 10^{-6}$, two-tailed Wilcoxon rank-sum test, Fig. 7k). These results indicate that aDS DA release reflects putative negative prediction errors in dips to extinction cues and unrewarded ITI licks, which are opposite in polarity and precede the average latency of ACh late peaks in the same region.

### Changes in glutamate release onto aDS CINs cannot solely account for ACh increases during extinction

Next, we asked whether emerging ACh increases to cues after partial extinction may develop as a consequence of increases in excitatory glutamatergic drive. To test this possibility, we expressed the genetically encoded glutamate sensor iGluSnFr[67] selectively in aDS cholinergic interneurons of ChAT-cre mice to measure rapid changes

in glutamate release (GluCIN) during partial extinction (Fig. 8a–c). Interestingly, GluCIN release in the aDS to the light stimulus following Pavlovian learning was bi-phasic, consisting of an initial fast latency increase followed by a slower latency increase (Fig. 8d, e). Consistent with the relative absence of strong ACh release peaks in the aDS to the conditioned tone (Figures S1d, S2d,f), no rapid changes in aDS GluCIN release were present for the tone, despite similar pre-licking behavior and velocity changes (Fig. 8e). This observation ruled out potential contributions of movement changes to the fast bi-phasic profile in the aDS GluCIN signal. Following light cue partial extinction (from 100 to 20% reward probability), there was no significant change in this bi-phasic profile (Fig. 8e, early peak: $p = 0.96$, two-tailed Wilcoxon rank-sum test; late peak: $p = 0.34$, two-tailed Wilcoxon rank-sum test) indicating that an increase in glutamate release onto the CINs was not responsible for the increase of aDS ACh release during partial extinction (Fig. 4). Despite this, the latency of the slower component of the GluCIN release aligned, on average, with the latency of the ACh late peak increase following extinction (median latency to Glu late peak = 0.64 s, *vs* median latency to ACh late peak = 0.8 s), indicating that the influence of this input may be

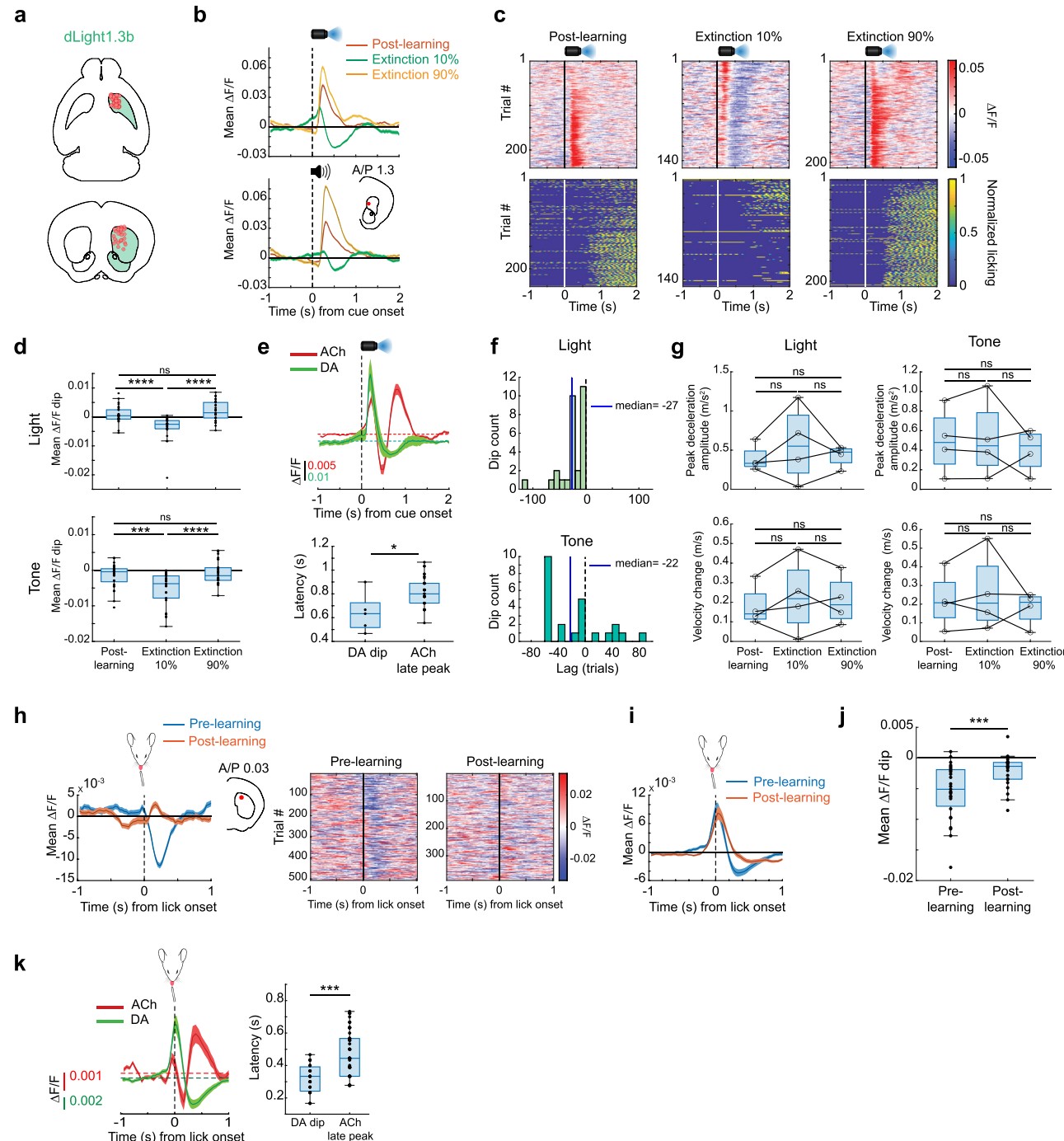

'unmasked' to drive the ACh increase, perhaps by disinhibition through emergent DA dips (see Discussion).

## Discussion

We applied high density arrays of small diameter optical fibers to map the spatiotemporal dynamics of rapid changes in ACh release across the striatum during Pavlovian learning and partial extinction in behaving mice. Our measurements reveal a topographic organization in the evolution of ACh release across timescales ranging from 100 s of milliseconds to days as Pavlovian associations are learned then broken. Most notably, we identified a specific region of the anterior dorsal striatum (aDS) with changes in ACh release over learning and extinction consistent with positive and negative RPE encoding for cues, rewards, and unrewarded consummatory actions. Changes were inverse in polarity to those widely reported for DA release and RPE

encoding was preferentially expressed in long latency components (dips and late peaks) of the multi-phasic ACh release profiles. Some changes in early peaks consistent with positive RPE encoding similar to DA were observed[22,39] but were more sparse, concentrated ventrally, and primarily restricted to the cue period. We hypothesize that positive prediction error encoding in ACh dips defines plasticity windows over time and space which facilitate initial formation of Pavlovian associations. In contrast, negative prediction error encoding in ACh late peak elevations may play an active role in erasing learned associations which are no longer valid, resulting in down-regulated behavioral responding to previously valued cues. These signals may support a general role for striatal ACh in forms of behavioral flexibility identified in previous manipulation studies. While our study identified the aDS as a potential locus for downshifting learned cue-reward associations, different forms of flexibility may recruit changes in ACh

**Fig. 7 | Dips in dopamine release in the anterior dorsal striatum are present for unrewarded lick onsets before learning and for cues after extinction.**
**a** Schematic showing the locations of fibers ($n = 29$ fibers in 4 mice) in the aDS for measurements of DA release with dLight1.3b. **b** Light (top) and tone (bottom) cue onset triggered averages of ΔF/F for a single fiber (location in the coronal plane in inset) for trials where cues were associated with 100% (post learning phase), 90%, and 10% (extinction) probabilities. Shaded region, S.E.M. **c** Light cue-aligned ΔF/F (top) and licking (bottom) for all trials for each phase included in the triggered averages in (**b** top). **d** Box plots showing the median DA ΔF/F minima (dip, see "Methods") for light (top) and tone (bottom) cue trials for trials post learning and the 10 and 90% reward probability phases during partial extinction. Each dot is a single fiber ($n = 29$ fibers in 4 mice). Box plots with whiskers extending from the minimum to the maximum values. The center line represents the median, the box bounds show the 25th and 75th percentiles. Data were analyzed using a linear mixed-effects model with mouse as a random effect (light cue: $F_{(2,84)} = 27.21$, $p < 0.0001$; tone cue: $F_{(2,82)} = 17.27$, $p < 0.0001$). Bonferroni post hoc comparisons: for light, **** $p < 0.0001$ post learning vs extinction 10%, n.s, $p = 0.43$ post learning vs extinction 90%, **** $p < 0.0001$ extinction 10% vs 90%, for tone, *** $p = 0.00024$ post learning vs extinction 10%, n.s, $p = 1.00$ post learning vs extinction 90%, **** $p = 0.00001$ extinction 10% vs 90%. n.s.: not significant. **e** Top: mean light cue onset triggered DA ($n = 29$ fibers across 4 mice) and ACh ($n = 68$ aDS fibers across 8 mice) ΔF/F for cue onset during extinction (10% and 20% for DA and ACh, respectively). Shaded region, S.E.M. Bottom: Box plot of latencies to the minimum DA (DA dip) or maximum (ACh late peak) for all fibers in DA or ACh sensor-expressing mice, respectively, in the extinction phase. Each dot is one fiber. Box plots show the median (center line), the 25th and 75th percentiles (bounds of the box), and the minimum and maximum values (whiskers). Two-tailed Wilcoxon rank-sum test, $W_{(5,53)} = 64$, * $p = 0.0162$ (two-tailed). **f** Histogram showing the latencies across fibers of the emergence of significant DA dips to light (top) and tone (bottom) cues relative to the decrease in lick index following high to low reward probability transitions. Blue vertical lines indicate the medians of the distributions. **g** Box and

whisker plots showing the peak decelerations and velocity changes across all mice ($n = 4$ mice) following light (left) and tone (right) cue onsets. Each datapoint is the mean for one mouse. Box plots indicate the median (center line), the 25th and 75th percentiles (bounds of the box), and the minimum and maximum values (wishers), n.s., not significant, Friedman test (two-tailed) with Dunn's multiple comparisons, $p > 0.05$. The exact $p$-values and additional statistical details can be found in Supplementary Table 1. **h** Left: Mean ΔF/F for a fiber (location in inset) aligned to the onset of spontaneous, unrewarded spout licking bouts in the ITI on trials pre (blue) and post (orange) initial Pavlovian learning. Shaded region, S.E.M. Right: Lick bout-aligned ΔF/F for all bout onsets included in triggered average on left. **i** Mean lick bout onset-triggered ΔF/F across all mice and fibers ($n = 29$ fibers across 4 mice) for bouts pre (blue) and post (orange) learning. Shaded region, S.E.M. **j** Box plots as in (**d**) showing the median DA ΔF/F minima for spontaneous lick bout onsets pre- and post-learning. The box plots indicate the median (center line), 25th and 75th percentiles (bounds of the box), and the minimum and maximum values (whiskers). Each dot is one fiber (27 fibers for pre learning, 24 fibers for post learning, across 4 mice). Data were analyzed using a linear mixed-effects model with mouse as a random effect ($F_{(1,49)} = 12.73$, *** $p = 0.00082$). **k** Left: Mean lick bout onset-triggered DA ($n = 29$ fibers across 4 mice) and ACh ($n = 68$ aDS fibers across 8 mice) ΔF/F in the aDS (see "Methods") for bouts pre learning. Shaded region, S.E.M. Right: box plot of latencies to the minimum (DA dip) or maximum (ACh peak) for all fibers in DA or ACh sensor-expressing mice respectively in the pre learning phase. The box plot indicates the median (center line), 25th and 75th percentiles (bounds of the box), and the minimum and maximum values (whiskers). Each dot is one fiber. $W_{(27,68)} = 734$, *** $p < 0.001$, Two-tailed Wilcoxon rank-sum test. Brain schematics in (**a**), (**b**) and (**h**) were adapted from the Allen Mouse Brain Common Coordinate Framework (CCFv3) (https://atlas.brain-map.org/. Mouse head schematic in (**h**), (**i**) and (**k**) adapted from SciDraw (scidraw.io), licensed under CC BY 4.0 (https://creativecommons.org/licenses/by/4.0/). Source https://doi.org/10.5281/zenodo.3925903). Source data are provided as a Source data file.

in distinct striatal regions. Indeed, one study observed a correlation between ACh transients in the medial nucleus accumbens shell and cocaine place preference extinction[29] and another observed ACh increases during extinction of lever pressing behavior in the dorsal medial striatum[33]. Further, ACh manipulations can result in diverse, sometimes conflicting, effects across task paradigms and regions. Future surveys of large-scale ACh release will be necessary to clarify the specific ACh dynamics across striatal regions associated with different forms of behavioral flexibility.

Recordings of striatal CIN firing and ACh release have largely focused on changes occurring during initial associative learning. These studies have primarily reported peak and dip components to predictive cues and rewards only partially consistent with positive RPE encoding[8,10–12,15,22]. One study reported positive RPE encoding in elevations in CIN firing in the ventral, but not the dorsal striatum[22]. Our results show that changes to cues and rewards are topographically organized across distinct striatal axes for different signal components and that full positive RPE encoding is present predominantly for the dip component, is cue modality specific, and is concentrated in the aDS. These findings indicate that discrepancies across prior studies may be due, at least in part, to limited spatial sampling and that the three components of the multi-phasic ACh release profile (early peak, dip, late peak) are shaped by different underlying mechanisms or inputs, which vary across striatal regions. Several studies provide evidence that ACh dips are enhanced by (though not dependent on) inhibitory DA D2 receptor signaling on CINs[11,12,59], which may contribute to the positive RPE encoding inverse to DA in aDS dips. Recordings of midbrain DA neurons have reported full RPE encoding to conditioned cues and rewards across a large majority of the recorded population (particularly in VTA[35,37,54]). However, some regional differences in striatal DA release (and positive RPE encoding) have been observed[68–71], so it is possible that spatial variations in positive DA RPE signaling drive localized inverse RPE encoding in ACh release. Alternatively (or in addition), DA may exert a different relative influence in CIN activity across regions[64].

What are the possible consequences of the spatially organized encoding of positive RPEs? ACh dips have been suggested to open a plasticity window to permit DA elevations to drive potentiation of excitatory synapses onto D1 receptor expressing projection neurons during associative learning[3,20,21,72,73]. Our results suggest that ACh dips may gate plasticity as a function of learning selectively within specific regions of the dorsal striatum. Changes in both peaks and dips were expressed across partly non-overlapping regions for tone and light cues (Fig. 2m, and S2g), suggesting that plasticity gating may be sensory modality specific across regions, perhaps aligning with the topography of glutamatergic inputs to striatal CINs[52,53,64]. We propose that regional variations in ACh release shape where, when, and how synaptic plasticity is expressed in striatal sub-circuits and cell types.

As cues lost their reward predictive value during extinction, we observed an elevation of late peaks concentrated in the aDS, the same region in which positive RPE encoding was observed in ACh dips (Figs. 4 and S4). Late peak changes were consistent with negative RPE encoding, representing the difference between the previously learned value of the predictive cue and the lower predictive value after extinction. Changes over learning also occurred in late ACh peaks to spontaneous unrewarded ITI licks, consistent with negative RPEs and partially overlapping with the aDS negative RPE encoding at cue (Fig. 5 and S6). These results suggest that putative positive and negative RPEs are encoded preferentially in the aDS in the dip and late peak components respectively. How might the negative RPE encoding emerge in ACh late peaks? Our evidence suggests that DA dips in the aDS may be converted to ACh peak elevations via the cessation of a tonic D2 receptor mediated inhibition of CINs. Consistent with this idea, optogenetic inhibition of DA release is capable of rapidly (<200 ms) elevating ACh release in-vivo, an effect which depends on D2 receptors[59]. The timing of this effect is consistent with the average latency of ACh peaks relative to the DA dips (Fig. 7, albeit across separate groups of mice). Our glutamate release measurements onto CINs indicated a long-latency component of excitatory glutamate release which does not increase during extinction, but which may be

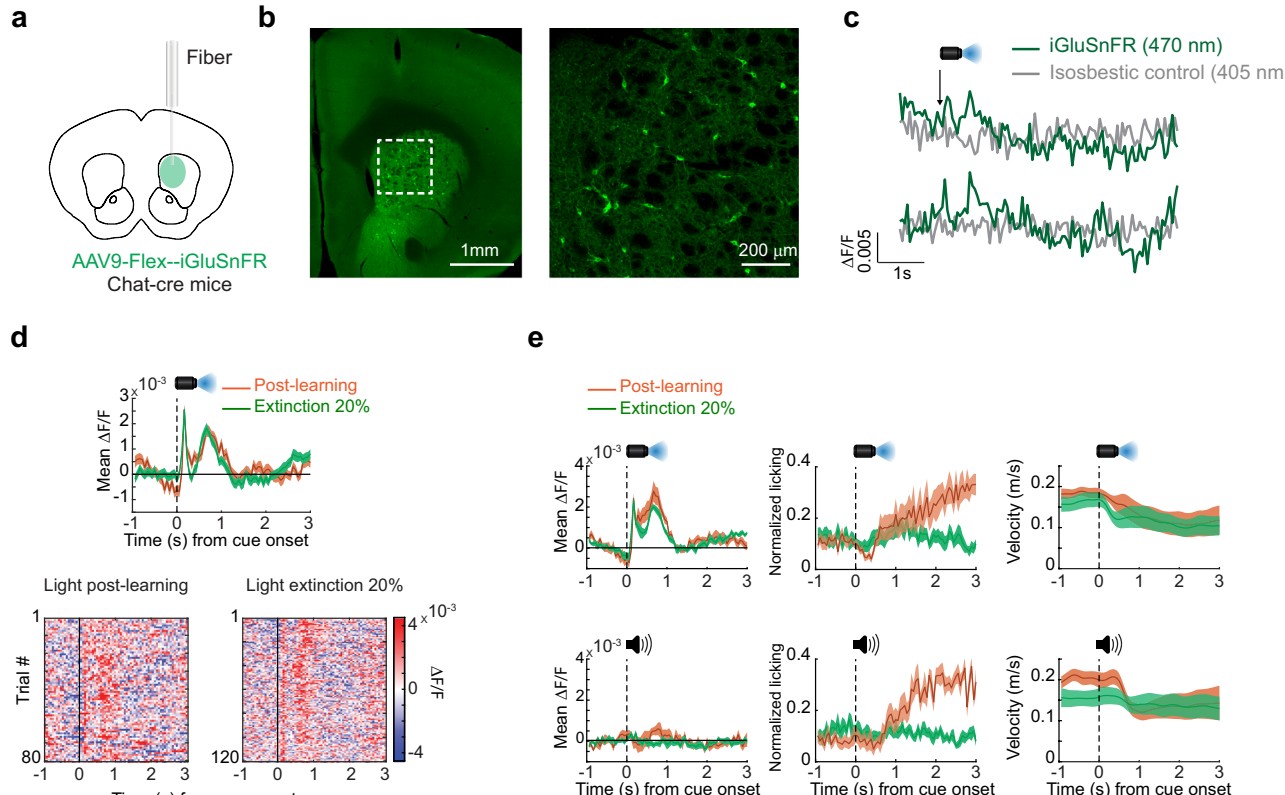

**Fig. 8 | Cue-evoked glutamate release onto anterior striatum cholinergic interneurons does not change during extinction. a** Schematic showing strategy for single fiber optical measurements of bulk glutamate release selectively on aDS CINs with iGluSnFR in ChAT-cre mice. **b** Left: fluorescence image of a coronal section of the anterior striatum showing iGluSnFR expression. Right: zoomed image of the region in the white box on left showing restricted expression to cholinergic interneurons. Experiments were independently repeated n = 2 times with similar results. **c** Representative ΔF/F traces aligned to light cue onset for two trials in one mouse. **d** Top: mean ΔF/F from one representative fiber from a single mouse aligned to light cue onset for trials post learning (orange) and partial extinction (20% reward probability cue, green). Shaded regions, S.E.M. Bottom: light cue-aligned ΔF/F for all trials included in the triggered averages at top. **e** Mean ΔF/F (left), licking (middle), and velocity (right) (n = 8 sessions across 2 fibers in 2 mice) aligned to light (top) and tone (bottom) cue onsets for post learning and 20% partial extinction phases. Shaded regions, S.E.M. Brain schematic in (**a**) was adapted from the Allen Mouse Brain Common Coordinate Framework (CCFv3) https://atlas.brain-map.org/.

'unmasked' by the emergent DA dips (Fig. 8). These results implicate a conjunction of appropriately timed glutamate input and disinhibitory DA dips to drive an emergence of ACh peaks encoding negative RPEs. This may occur either through post-synaptic plasticity of glutamatergic inputs onto CINs[18], or through immediate effects of the DA dips on elevating CIN excitability.

Why would the emergence of negative RPE encoding in ACh late peaks be spatially restricted to the aDS? One possibility is that the DA dips, encoding negative RPEs, which disinhibit the CINs are also restricted to aDS. Variations in negative RPE encoding have been observed across DA neurons and in DA release, though region specific DA recordings have not localized dips exclusively to the aDS[37,69,71]. A second possibility is that the glutamatergic input unmasked by the DA dip is aDS specific. In support of this, significant DA dips were present for both the tone and light cues in the aDS, but late peaks in glutamate release were selective for the light. Similarly, ACh late peaks for the tone were not as prominent in the aDS and were shifted posterior and lateral relative to the light (Fig. 4l and S4c), indicating that distinct patterns of glutamate release may shape the regional topography of negative RPE encoding for different cue modalities. Third, DA may exert a stronger inhibitory influence on ACh in the aDS than in other regions. In support of this, studies in brain slices have observed a gradient of D2 mediated inhibition of CINs in the striatum, with particularly strong suppression in the aDS, and an opposite effect in the ventral striatum[64]. Simultaneous, striatum wide measurements of DA and ACh during learning and extinction will be needed to further investigate these potential mechanisms.

Extinction has been proposed to occur both through an 'unlearning' process involving reversal of plasticity occurring during associative acquisition and through new inhibitory learning[63,74]. The ACh peaks in the aDS are well positioned to promote both of these processes in parallel through their differential effects of synaptic plasticity on striatal projection neuron cell types. Pavlovian associations are believed to involve the strengthening of synapses onto D1 receptor expressing direct pathway neurons (dSPNs) driven by phasic elevations in DA to unpredicted rewards and reward-associated cues (e.g. positive RPEs)[21,54,72]. Studies in-vitro have found that M4 muscarinic ACh receptors, expressed preferentially on direct pathway spiny projection neurons (dSPNs), promote long term synaptic depression[75,76]. Thus, the increases in cue evoked ACh during extinction could weaken the previously potentiated synapses onto dSPNs. Importantly, while DA dips encoding negative RPEs have been proposed to contribute to extinction, it is unclear how DA dips alone could weaken previously potentiated dSPN synapses because low affinity D1 receptors are not strongly activated by basal DA levels and therefore are not sensitive to DA dips[77]. In contrast, CINs express the high affinity inhibitory D2 DA receptor, so DA dips promote rapid ACh increases through disinhibition[59,77]. Therefore, disinhibition of CINs may be a mechanism to convert negative prediction errors encoded by DA dips into ACh signals that can drive de-potentiation of dSPN synapses and 'unlearning' of invalid cue-reward associations. In parallel, the aDS ACh elevations (and DA dips) during extinction may elevate excitability (relative to dSPNs) and strengthen synapses of D2 expressing indirect

pathway neurons (iSPNs) via M1 receptors[3,4,78]. This may result in new inhibitory learning, where iSPNs become more strongly activated to the extinction cue to actively suppress appetitive responding.

## Methods

All details of key lab materials used and generated in this study are listed in a Key Resources Table on Zenodo (https://doi.org/10.5281/zenodo.14728851). Protocols associated with this work can be found on protocols.io (https://doi.org/10.17504/protocols.io.n2bvjdx65vk5/v1).

### Mice

For all experiments adult male and female (n = 33 mice in total, postnatal 3-5 months, 24–30 g), wild type (WT) C57BL6 /J (n = 15, 10 males and 5 females, Jackson Labs, strain #00664) and heterozygous ChAT-cre mice (n = 18, 10 males and 8 females, ChAT-IRES-Cre, Jackson Labs, strain# 006410) were used in this study. Mice were initially housed in groups and then individually housed following surgery, under standard laboratory conditions (20–26 °C, 30–70% humidity; reverse 12-h/12 h light/dark cycle; light on at 9 p.m.) with *ad libitum* access to food and water, except during water scheduling. The number of mice used for each experiment and analysis is indicated in figure legends and main text. Experiments were conducted during the dark cycle. All animal care and experimental procedures were performed in accordance with protocols approved by the Boston University Institutional Animal Care and Use Committee (protocol no. 201800554) and in compliance with the Guide for Animal Care and Use of Laboratory Animals.

### Multi-fiber array fabrication, calibration, and micro-CT scanning for fiber localization

**Array fabrication.** The multifiber arrays to enable large-scale measurements of ACh (and DA) release across the 3D volume of the striatum were fabricated in-house as previously described[44] (Fig. 1a). Briefly, 51–99 optical fibers (50 μm, 46 μm core and 4 μm cladding with 0.66NA, Fiber Optics Tech) were threaded into holes (55–60 μm diameter) within a custom 3D-printed grid (3 mm W x 5 mm L, Boston Micro Fabrication) and secured with UV glue (Norland Optical Adhesive 61). Fibers were fixed at preset lengths from the bottom of the grid (measured under a dissection scope) and neighboring fibers were separated by at least 250 μm axially and 220 μm radially to target different locations throughout the striatum volume, ensuring no (or very minimal) overlap between fluorescence collection volumes[46,47,79]. Fiber ends on the non-implanted side were bundled and glued inside a polyimide tube (0.92 mm ID, MicroLumen) and then cut with a sharp razor blade to a ~1 cm length. The bundled fibers were then polished first on fine 6 μm grained polishing paper followed by 3 μm (ThorLabs) to create a uniform and smooth imaging surface and enable efficient light transmission through each fiber (Fig. 1a).

**Fiber localization.** Following the array fabrication, a calibration procedure was performed to match individual fibers on the implanted side to their corresponding locations on the bundle surface as previously described[44]. At the end of neuronal recordings and behavioral experiments, mice were injected intraperitoneally with Euthasol (400 mg kg-1, Covertus Euthanasia III) then perfused intracardially with phosphate-buffered saline (PBS 1%, Fisher Scientific), followed by paraformaldehyde (4% in 1% PBS, Fisher Scientific). The mice were decapitated, and the ventral side of the skull removed to expose the ventral side of the brain. The intact implanted brains were post-fixed for 24 h in 4% paraformaldehyde, rinsed with 1% PBS then transferred to a Lugol iodine solution (Carolina Scientific) diluted 1:3 in distilled water for 4 to 6 days. The diffusion of the Lugol solution to the brain enhances tissue contrast for computerized tomography scanning to enable fiber localization. The CT scanning and fiber localization were previously described[44]. Briefly, the 3-D CT scans of the intact implanted brains were registered to the The Allen Mouse Brain Common

Coordinate Framework atlas via a semi-automated landmark-based approach[48,80]. The fibers, bright in the CT scan, were then automatically identified via intensity thresholding, and the recording locations (ventral-most point) were mapped to their corresponding locations on the implanted grid and subsequently to their locations on the imaging surface. In addition to bringing all mice into a common coordinate space, registration with the atlas also enabled the recording locations to be automatically assigned atlas anatomical labels, which were further manually verified (Fig. 1d).

### Stereotaxic viral injections and chronic optical fiber implants

Mice were anesthetized with isoflurane (3–4%) and placed in a stereotaxic frame (Kopf instruments) on an electric heating pad (Physitemp instruments) and administered buprenorphine extended release for pre-operative analgesia (3.25 mg kg-1 subcutaneous, Ethiqa XR). Following induction, isoflurane was held at 1–2% (in 0.8–1 L min$^{-1}$ pure oxygen) and body temperature maintained at 37 °C, throughout the surgical procedure. The scalp was shaved and cleaned with iodine solution prior to exposing the skull. For experiments to record extracellular acetylcholine release using multi-fiber arrays, a large craniotomy was performed over the right hemisphere with a surgical drill (Midwest Tradition 790044, Avtec Dental RMWT) to expose the brain surface from −2.3 to 2 mm in the anterior-posterior (AP) direction and from 0.49 to 3.4 mm in the medial-lateral (ML) direction relative to bregma. AAV9-hSyn-ACh3.0 (WZ Biosciences[45]), 2.07 x $10^{13}$ GC ml$^{-1}$ diluted 1:2 in PBS was pressure-injected into the striatum of WT mice (n = 8) through a pulled glass pipette (tip diameter 30-50 μm) at 20-40 separate striatum locations chosen to maximize expression around fiber tips (200 nl at each location at a rate of 100 nl/min). For control experiments, a 1:2 mixture of AAV9-hysn-ACh3.0-mut[45] (WZ Biosciences), 2.54 x $10^{13}$ GC ml$^{-1}$ was injected into the striatum of WT mice (n = 3) using the same strategy. For experiments to record extracellular DA release using multi-fiber photometry, AAV5-CAG-dlight1.3b[66] (Addgene, # 111067), 1.7x x$10^{13}$ GC ml$^{-1}$ diluted 1:3 in PBS was injected into the striatum of WT mice (n = 4) at 10–40 total locations (200-800 nl at each location) using the same procedure. Following injections, the multi-fiber array was mounted onto the stereotaxic manipulator, the dura gently removed, and the array slowly lowered into position. The craniotomy was sealed with a thin layer of Kwik-Sil (WPI), and the array was secured to the skull surface using Metabond (Parkell). To allow head fixation, a metal head plate and ring (Atlas Tool and Die Works) were next secured to the skull with Metabond, and the implant surface was covered with a mixture of Metabond and carbon powder (Sigma Aldrich) to reduce optical artifacts. The fiber bundle was protected by a cylindrical plastic tube, extending ~1–2 mm above the fiber bundle, and secured around the bundle using a mixture of Metabond and carbon powder.

To drive the suppression of ACh release from cholinergic interneurons with tetanus toxin light chain (TelC, Fig. 6), small, circular craniotomies were drilled bilaterally above the injection sites (from bregma, in mm; AP: 1, ML: ± 1.4). Then either pAAV2/8-hSyn-FLEX-TeLC-P2A-EYFP-WPRE (Addgene, #135391[81], a gift from Bernando Sabatini's Lab, 5.14 x $10^{13}$ GC ml$^{-1}$) or ssAAV-5/2-hSyn1-dlox-TeTxL-C_2A_NLS_dTomato(rev)-dlox-WRPE-hGHp (Viral Vector Facility University of Zurich, 4.1 × $10^{12}$ VG ml$^{-1}$) diluted 1:1 in PBS was bilaterally injected in the anterior dorsal medial striatum of ChAT-cre mice (n = 6) at 4–12 sites per hemisphere (300 nl/site at a rate of 100 nl/min) at the following coordinates in mm; AP: 0.8, ML: ± 1.25, DV: -2.5 and -3; AP:1, ML: ±1.4, DV:-2.75 and -3. Control ChAT-cre mice (n = 8 Chat-cre mice) were injected with saline using the same strategy. To validate that TelC expression in the aDS leads to a reduction in ACh release, two ChAT-cre mice were bilaterally injected in the aDS using the same coordinates with ssAAV-5/2-hSyn1-dlox-TeTxLC_2A_NLS_dTomato(rev)-dlox-WRPE-hGHp diluted 1:1 in PBS. Additionally, the left hemisphere of each mouse was co-injected with AAV9-hSyn-ACh3.0 diluted 1:2 in PBS.

Following the injections, the craniotomies were sealed with Kwik-Sil (WPI), and the skull was sealed with Metabond (Parkell) and a metal head plate.

To measure extracellular glutamate release into cholinergic interneurons (Fig. 8), craniotomies were drilled above the injection sites in the right hemisphere (from bregma, in mm; AP: 1, ML: 1.4). A 1:1 mixture in PBS of the genetically encoded glutamate sensor AAV9.h-Syn-FLEX.8F-iGluSnFR.A184S[67] (Adgene, #106174), $1.8 \times 10^{13}$ GC ml$^{-1}$ was injected in aDS of ChAT cre mice ($n = 2$) at 6 sites (300 nl/site at a rate of 100 nl/min) at the following coordinates in mm: AP: 0.8, ML: 1.5, DV: −2.75, −3.25 and −3.75; AP: 1.1, ML: 1.5, DV: −2.75, −3.25 and −3.75. Then, a 100 μm core diameter optical fiber (MFC_100/125- 0.37NA) attached to a zirconia ferrule (Doric) was slowly lowered into the medial region of the aDS (AP:1, ML:1.4) to a final depth of 3 mm from bregma. The craniotomies were sealed with Kwik-Sil (WPI), the optical fiber and a head plate were secured to the skull with Metabond (Parkell). After the surgeries, mice were placed in a cage with a heating pad and received postoperative injections of meloxicam (5 mg kg$^{-1}$ subcutaneous, Covertus) and 1 mL of saline per day subcutaneously for 4 days after surgery. Mice were individually housed and allowed to recover in their cages for at least 2 weeks after surgery.

### Pavlovian conditioning task and behavior setup

Behavioral apparatus: One week before starting the Pavlovian conditioning training and photometry recordings, mice were placed on a water schedule, receiving 1 ml of water per day, and were maintained at 80–85% of their initial body weight for the duration of the experiments. Three to four days prior to training, mice were habituated to head fixation on the spherical styrofoam treadmill[51] (Smoothfoam, 8in diameter, Fig. 1b). The behavioral setup has been described in detail previously[44]. Briefly, mice were free to locomote on the treadmill during all experiments and the ball rotation in pitch, yaw, and roll directions was measured using optical computer mice (Logitech G203) through an acquisition board (NIDAQ, PCIe 6343). Water rewards (5 μL/reward) were dispensed through a water spout operated by an electronically controlled solenoid valve (#161T012, Neptune Research), mounted on a post a few mm away from the mice's mouths. Tongue spout contacts (a proxy for licking) were monitored by a capacitive touch circuit connected to the spout and confirmed with live video taken from a camera positioned to capture orofacial movements (Blackfly S USB3, BFS-U3-16S2M-CS, Teledyne Flir).

Pavlovian conditioning and partial extinction: Approximately three weeks post-implantation, mice began training on a dual cue delay Pavlovian conditioning task (Figs. 1b, 2a and 4a). In each session (one session/day), mice received 40 presentations of two different cues in a pseudorandom order: light and tone (20 presentations of each). Light cues were presented via a LED (Thor labs, M470L3, 470 nm) calibrated to deliver light at 7 mW intensity and mounted on a post holder ~20 cm away from the mouse, positioned 45 degrees contralateral to the implanted side. Tone cues (12 kHz, 80 dB) were presented via a USB speaker placed ~30 cm from the mouse. Each cue was presented for 6 s and was paired with a water reward (5 μL) delivered with 100% probability after a fixed 3 s delay from the cue onset. An ITI was randomly drawn from a uniform distribution of 4–40 s. A total of eight random non-contingent rewards per session were delivered during the ITI periods. The mice were trained for 7–12 consecutive days until they learned that both light and tone cues were associated with the delivery of a water reward (as measured by the lick index, see below). They were then trained for an additional 2–6 days. Following initial learning, mice were submitted to a partial extinction phase (Fig. 4a) in which the reward probability associated with one of the two cues was downshifted to 20% (10% for DA experiments, Fig. 7), and the other cue to 80% (90% for DA experiments). During the extinction phase, mice received 60 presentations of the two cues (30 presentations of each). Training continued until mice showed significantly diminished pre-

licking for the 20% cue relative to the 80% for 4−7 sessions, then cue probabilities were reversed. The order of light and tone cue probabilities was counterbalanced across mice. For TelC experiments, TelC and control mice were trained for a maximum of 8 sessions for each extinction phase.

### Multi-fiber photometry recordings

Fluorescence measurements from the multi-fiber arrays were conducted using a custom built microscope (Fig.1b) mounted on a 4' W x 8' L x 12' thick vibration isolation table (Newport). Details of the microscope were described previously[44]. Excitation light for the fluorescent sensors (ACh3.0, ACh4.3 mut, iGluSnFR and dLight 1.3b) was provided by two high power LEDs (470 nm and 405 nm; Thor labs, No. SOLIS-470C, SOLIS-405C). Excitation light was bandpass filtered (Chroma No. ET405/10 and ET473/24) then coupled into a liquid light guide (Newport No. 77632) with lenses ($f = 60$ mm and 30 mm, Thor labs No. LA1401-A and LA1805) and a collimating beam probe (Newport No. 76600). The liquid light guide was connected to a filter cube on the microscope, directing excitation light into the back aperture of the microscope objective (10x, 0.3NA, Olympus Model UPLFLN10X2) via a dichroic beam splitter (Chroma Model 59009bs). The light power at the focal plane of the objective was adjusted to be within the range of 80-85 mW, resulting in a power of 1.6 - 2 mW/mm$^2$ at the fiber tips[44]. Emission light was bandpass filtered (Chroma, No 525/50 m) and focused with a tube lens (Thor labs, No TTL165-A) onto the CMOS sensor of the camera (Hamamatsu, Orca Fusion BT Gen III), creating an image of the fiber bundle (Fig. 1a). To enable precise manual focusing, the microscope was connected to a micromanipulator (Newport Model 96067-XYZ-R) and mounted on a rotatable arm extending over the head-fixation setup to facilitate positioning of the objective above the imaging surface over the mouse head. Imaging data acquisition was performed using HCImage live (HCImage live, Hamamatsu). Single wavelength excitation was carried out with continuous imaging at 30 Hz (33.33 ms exposure time), via internal triggering. Dual wavelength excitation was performed in a quasi-simultaneous externally triggered imaging mode, where the two LEDs were alternated and synchronized with imaging acquisition via 5 V digital TTL pulses. 470 nm excitation was alternated with 405 nm excitation at either 36 Hz (20 ms exposure time) or 22 Hz (33.33 ms exposure time) to achieve a frame rate of 18 Hz or 11 Hz for each excitation wavelength, respectively. Recordings acquired at different sampling rates were downsampled or upsampled using a 1-D interpolation with Matlab's *interp1* function using the spline method. A custom MATLAB software controlled the timing and duration of TTL pulses through a programmable digital acquisition card (NIDAQ, National Instruments PCIe 6343). Voltage pulses were transmitted to the NIDAQ from the camera following the exposure of each frame to confirm proper camera triggering and to synchronize imaging data with behavior data.

### Statistics and reproducibility

Data were processed and analyzed using built-in and custom functions in Matlab (Matworks, version 2020b, 2022b and 2023a), Python, or GraphPad Prism10 (GraphPad Software). Some fibers were excluded from analysis based on localization outside the striatum or poor signal-to-noise ratio (see below for details). Exclusion was performed prior to any statistical analysis of task related signals. Tests for significance are indicated in the text and figure legends. Sample sizes were chosen to effectively measure experimental parameters while remaining in compliance with ethical standards to minimize animal usage. There was no randomization or blinding conducted.

### Multi-fiber photometry signal preprocessing

The acquired time series videos of the fiber bundles were first motion-corrected using a whole-frame cross-correlation algorithm described previously[51,82] then visually inspected to confirm post-correction image

stability. Circular regions of interest (ROI, ~ 25 μm diameter) were manually selected for each fiber. The resulting set of ROIs comprised a mouse-specific ROI template, which was then fit and applied to each subsequent imaging video, enabling the identity of each ROI to remain consistent across multiple recording sessions. To determine the change in fluorescence ΔF/F, the mean fluorescence extracted from each ROI was normalized to a baseline, which was defined as the 8th percentile fluorescence over a 30-s sliding window[44]. To remove low frequency artifacts, the ΔF/F signals were high-pass filtered using a finite impulse response filter with a passband frequency set at 0.3 Hz. This frequency was determined based on the observed differences in the dynamics of the ACh signal compared to the control signals (ACh-mut and the isosbestic 405 nm LED signal, Fig. 1h, i). Most analyses were conducted on non z-scored or peak normalized ΔF/F values in order to identify relative differences in signal magnitude across task and training phases. Changes in overall signal magnitude over training due to changes in sensor expression or fiber collection efficiency were accounted for by examining the stability of signals to unpredicted water reward for each fiber (Fig. 1f).

## Quantification and statistical analysis

**Definition of learning phases.** Lick indices were computed for each trial and session to assess learning of the Pavlovian associations. The lick index was defined by the following formula:

$$Lick\ index = \max(0, \log(anticipatory\ lick) \times ((anticipatory\ lick - lick\ ITI) \div (anticipatory\ lick + lick\ ITI))$$

where anticipatory lick is the sum of lick spout counts across a 1 s window before reward delivery, and lick ITI is the sum of lick count across a 1 s window before cue onset. To determine the learning phases (pre- and post-learning), the mean lick indices across all trials of each acquisition session were compared to those of the first acquisition session using a two tailed Wilcoxon rank-sum test. A p-value < 0.05 was considered statistically significant. For initial Pavlovian conditioning, sessions where the mean lick indices for both cues were not significantly different from the first session were considered pre-learning sessions (2.9 ± 0.62), while sessions where mean lick indices of both cues were significantly higher than the first session were considered post-learning sessions (3.27 ± 0.5) (Fig. 2c, d). Sessions where only one cue had a significantly elevated lick index were omitted from analysis (2.90 ± 0.75 sessions). For the extinction phase (Fig. 4), sessions were included in analyses if the lick indices of the 20% probability cue were significantly (Wilcoxon rank-sum test, p < 0.05) lower than lick indices for the same cue on the last day of the post-learning phase or the preceding 80% probability session (for reversal sessions). The fraction of time spent licking during the cue period (Fig. 2d) was calculated across the entire 3-s window after cue onset, and the fraction of time spent licking during the ITI period was calculated across the entire ITI period, excluding 0.5 s before cue onset, 3 s after cue offset, and 6 s after any unpredicted reward delivery. Spontaneous lick bout onsets in the ITI (Figs. 5 and 7) were defined as the first lick of lick bouts that are not preceded by any licking for at least two seconds.

For lick analyses in TelC experiments (Fig. 6), all the extinction sessions (20% reward probability for both light and tone cues combined) were compared with sessions of pre- and post-learning (Two-way ANOVA followed by Tukey *post hoc* analysis to account for multiple comparisons within and between TelC and control mice, p < 0.05 was considered statistically significant).

**Relationships of signal with velocity and acceleration.** Analog signals from the optical mice were converted to m/s[44], and the pitch and roll were combined to compute a total velocity calculated

as: $\sqrt{pitch^2 + roll^2}$. The velocity traces were then smoothed using a Savitzky-Golay moving average filter with a moving window of 250 ms and a 2nd degree polynomial. Acceleration traces were derived from the smoothed velocity (acceleration = Δvelocity/Δtime), and further filtered using the same Savitzky-Golay moving average filter parameters.

To partially account for generalized relationships between treadmill locomotion and ΔF/F, the filtered fluorescence signals during ITI periods were fit to a generalized linear model (GLM) using smoothed linear velocity and acceleration as continuous predictors, each with positive and negative phases. The optimal positive and negative phase differences between ITI ΔF/F and velocity/acceleration were first identified through cross-correlations with a maximum lag of ±0.5 s. Next, correlation coefficients were calculated by fitting the phase-shifted velocities/accelerations to the ITI ΔF/F via least squares linear regression using Matlab's *fitglm* (equations below):

$$signal_{\Delta F/F_{filtered}} = \beta_0 + \beta_{v1} \times v_1 + \beta_{v2} \times v_2 + \beta_{a1} \times a_1 + \beta_{a2} \times a_2 + \varepsilon \quad (1)$$

GLM training, where $signal_{\Delta F/F_{filtered}}$ is the filtered ΔF/F of the ITI periods, $v_1/v_2$ and $a_1/a_2$ are positive/negative phase-shifted velocities and accelerations respectively, $\beta_{v1}/\beta_{v2}$ are the correlation coefficients of positive/negative phase-shifted velocities, $\beta_{a1}/\beta_{a2}$ are the correlation coefficients of positive/negative phase-shifted accelerations, $\varepsilon$ is the error term.

$$signal_{velocity} = \widehat{\beta_0} + \widehat{\beta_{v1}} \times v_1 + \widehat{\beta_{v2}} \times v_2 + \widehat{\beta_{a1}} \times a_1 + \widehat{\beta_{a2}} \times a_2 \quad (2)$$

$signal_{velocity}$ is the estimated contribution of velocity and acceleration to the ΔF/F signal, $v_1/v_2$ and $a_1/a_2$ are the same phase-shifted velocities and accelerations as described above, $\hat{\beta}$ are the estimated correlation coefficients generated by the GLM. Non-significant $\hat{\beta}$, i.e. with a p > 0.05, were set to 0 and were not included in the estimated corresponding velocity or acceleration contribution.

The final ΔF/F was computed by subtracting the velocity contribution ($signal_{velocity}$) from the filtered signal ($\Delta F/F_{filtered}$).

To further address the possibility that variations in treadmill velocity at cues or rewards contribute to ΔF/F signal changes, signals and velocity/acceleration were compared for different learning phases for each fiber (Supplementary Fig. S7). To identify decelerations occurring to cues, rewards, or lick bout onsets, deceleration periods were first defined in peri-event windows as consecutive bins with negative acceleration values. Peak deceleration (Supplementary Fig. S7a–c; Fig. 7g) was defined as the minimum value of acceleration during any single continuous deceleration period, in which the total velocity change exceeded a threshold of 8 cm/s. Total velocity changes following cue or reward (Supplementary Fig. S7a–c; Fig. 7g) were defined as the maximum change of velocity following the event onset, relative to pre-event. Total velocity change and peak decelerations were calculated from cue onset to 0.6 s after, and from −0.2 to 0.6 s relative to reward consumption (first spout lick after delivery). For non-contingent spontaneous licks during ITI periods, total velocity change and peak decelerations were calculated from −0.5 to 0.1 s after lick onset. Large and small deceleration trials were defined as trials where the peak deceleration was greater than the 80th percentile or less than the 20th percentile of the total peak deceleration distribution across all mice, respectively (Supplementary Fig. S7h–k). For analyzing the contribution of signals related to spontaneous, non-task related decelerations (Supplementary Fig. S7d–g), decelerations were first identified within ITI periods, excluding all times around spontaneous licks and unpredicted reward deliveries. Periods of continuous negative acceleration were identified and the troughs were included in triggered average analysis if they were lower than -1.5 m/s², and there was a total velocity change over the deceleration period of at least 20 cm/s. These thresholds were chosen conservatively to ensure that

ITI decelerations matched (or exceeded) the average deceleration magnitudes observed in the task (Supplementary Fig. S7a–c). ACh ΔF/F signals were aligned to the deceleration trough, and maximum and minimum values of the mean trough triggered average were calculated from a ±1 s window around the peak. Statistical significance of minima and maxima was determined by comparison to the 95% confidence intervals of a bootstrap-generated random sampling distribution (2000 iterations). To evaluate the potential influence of decelerations occurring at the cue or reward on learning-related changes, correlations (Pearson's) were calculated between significant ITI deceleration minima or maxima and the magnitude of the mean ΔF/F for different signal components during the cue and reward periods, limited to fibers with both significant deceleration and task-related changes for each component (Supplementary Fig S7l, m; and Supplementary table 2).

### Identification of multi-phasic ACh release components and changes with learning

Analysis was conducted on 295 fibers across the 3D volume of the striatum, out of a total of 505 implanted fibers, collected from 8 mice (with 37,36,47,44,45,31,29 and 26 fibers in each respective mouse). Additionally, analysis was performed on 110 fibers across the striatum, out of 185 implanted fibers, obtained from 3 mice expressing the non-functional ACh sensor (with 43, 44, and 23 fibers in each mouse respectively).

Significant positive or negative changes in ACh release were first identified around each event (cue onset, reward delivery, lick bout onset) within peri-event windows (0–1.5 s for cue, −0.5 –1.5 s for reward, and −0.5–1 s for lick onsets). Significance was determined as a signal change exceeding 3 standard deviations above (peaks) or below (dips) the mean of the ITI signal in a 1 s window prior to the event window. Local maximum (peaks) and minimum (dips) ΔF/F points were defined for each fiber and the timing of these significant points was plotted in histograms (Supplementary Figs. S1a, b; and Figure S6a). Based on the multi-phasic timing distributions from the histograms, windows were defined for each event to define significant early peak, dip, and late peak components (Supplementary Fig. S1a, b). The windows were as follows for early peak, dip, and late peak respectively: cue onset: 55 ms to 444 ms, 333 ms to 833 ms, 500 ms to 1222 ms; reward consumption: -200ms to 400 ms, 200 ms to 800 ms, 500 ms to 1200 ms; ITI lick bout onset: -270ms to 230 ms, 0 ms to 600 ms, 200 ms to 800 ms. For maps showing the presence of each set of components for each fiber, additional criteria were included that a signal must exceed 3 standard deviations relative to the ITI for at least 2 consecutive 0.056 ms time bins and reach a peak or trough of at least 0.005 ΔF/F or −0.005 ΔF/F respectively to isolate only the largest signal changes.

Differences in the mean event-evoked ΔF/F between different phases of learning for the three ACh signal components were calculated for each fiber by comparing the peak and dip values across single trials within each phase using an unpaired Wilcoxon rank sum test. P-values <0.01 were considered statistically significant, except for the difference in mean ITI lick bout-evoked ΔF/F, where p-values < 0.025 were considered significant. Signal changes consistent with RPE signaling were determined for each signal component and fiber across pairs of events (Figs. 3d,5g; and Figs S3e,S6f). Putative positive RPE encoding was defined as elevations in ΔF/F (larger peaks or deeper dips) with learning to the cue and decreases to the reward. Negative RPE encoding was defined as elevations to the cue with extinction and decreases to unrewarded ITI licks with learning.

To visualize the spatial distribution of learning related ACh changes across the striatum, ACh ΔF/F values for each fiber, representing the mean difference between phases of learning, were aggregated into smoothed heat maps (e.g., Figure 2j, k). First, mean ΔF/F difference values were binned into cubic arrays with an edge length of 0.05 mm based on the spatial location of each fiber. Fibers with no significant difference or no significant component were included as 0 s. Values

from fibers within each spatial bin were averaged then convolved with an exponentially decaying, distance-dependent spherical mask, with the decay rate calculated as: decay rate = $e^{-6 \cdot \text{euclidean distance}}$. The result was a 3-dimensional array of weighted sums for each spatial location, which was then reduced to 2-dimensional sagittal or axial plane plots by averaging values along the collapsed dimension. Regions with the largest signal changes (white contours) were identified as containing points with amplitudes above the 90th percentile of the corresponding reduced 2D array for increases and decreases separately.

### Spatial gradient analysis

To quantify the variation in signal changes with learning/extinction along each anatomical axis (Figs. 2l, m and 4k, l), a linear mixed-effects model was constructed which described the signal change amplitudes as a function of spatial coordinates along the anterior-posterior (AP), medial-lateral (ML), and dorsal-ventral (DV) axes. Spatial coordinates were included as fixed effects, while mouse identity was treated as a random effect to account for individual variability between subjects. The model is described by the following equation:

$$
\begin{aligned}
y_{ijk} = \beta_0 &+ \beta_{AP} \times AP_i + \beta_{ML} \times ML_j + \beta_{DV} \times DV_k + u_{AP,mouse} + u_{ML,mouse} \\
&+ u_{DV,mouse} + u_{mouse} + \epsilon_{ijk}
\end{aligned}
\tag{3}
$$

where $y_{ijk}$ is the amplitude of the ΔF/F differences, $\beta_0$ is the intercept, $\beta_{AP}$, $\beta_{ML}$, $\beta_{DV}$ are the fixed effect coefficients for the AP, ML and DV coordinates respectively, $u_{mouse}$, $u_{AP,mouse}$, $u_{ML,mouse}$, $u_{DV,mouse}$ are random effect intercept and coefficients for AP, ML and DV coordinates, for individual mice, and $\epsilon_{ijk}$ is the residual error term. $[\beta_{AP}, \beta_{ML}, \beta_{DV}]$ defined the striatal axes of maximal variation. The sign of the spatial coefficients was set so that the direction of the predominant signal change for each component was positive (e.g. larger dips or larger peaks were both positive), so that the sign of the coefficient indicated the direction of the spatial gradient (rather than the sign of the change) for comparison across components that changed in opposite directions.

To compare gradient coefficients between ACh components or experimental contexts (light vs tone for example), a similar model was fit, which included an interaction term between the coefficients of interest. This model is represented by the following equation:

$$
\begin{aligned}
y_{ijkl} = \beta_0 &+ \beta_{AP} \times AP_i + \beta_{ML} \times ML_j + \beta_{DV} \times DV_k + \beta_g \times Group_l \\
&+ \beta_{g,AP} \times Group_l \times AP_i + \beta_{g,ML} \times Group_l \times ML_j \\
&+ \beta_{g,DV} \times Group_l \times DV_k + u_{AP,mouse} + u_{ML,mouse} \\
&+ u_{DV,mouse} + u_{g,mouse} + u_{mouse} + \epsilon_{ijk}
\end{aligned}
\tag{4}
$$

In this model, $y_{ijkl}$ is the amplitude of the ΔF/F differences, constructed by stacking the two amplitude datasets for comparison, with their corresponding AP, ML and DV coordinates. $Group_l$ is a categorical variable distinguishing the two coefficients (e.g. light or tone) being compared, $\beta_g$ represents the coefficients of the categorical variable, $\beta_{g,AP}$, $\beta_{g,ML}$, $\beta_{g,DV}$ are the interaction coefficients between groups and AP, ML, and DV, $u_{g,mouse}$ are random effect coefficients for groups. All other parameters not explicitly mentioned are the same as the non-interaction model. Wald t-test was used for statistical significance, a p-value < 0.05 of the interaction coefficients i.e. $\beta_{g,AP}$, $\beta_{g,ML}$ and $\beta_{g,DV}$ is considered significant.

### Relative signal timing calculations

Latencies for ACh peaks or dips were determined as the time between the event onset and the time of the minimum or maximum signal amplitude in the triggered average for a given signal component (early/late peak, dip). To determine the relative timing between the increase in light and tone cue-evoked late peak ΔF/F and the subsequent decrease in lick index following partial extinction (Figs. 4m–p

and 7f), lick indices and late peak amplitude values for each fiber across trials were smoothed using Matlab's *smooth* function with the lowess method (local regression using weighted linear least squares and a first-degree polynomial model with a moving window of 30 trials). Next, Matlab's *cusum* function with a climit of 5 and mshift of 2 was used to identify the change point for lick indices and late peak $\Delta F/F$ (a decrease in lick index and an increase in late peak). The *cusum* algorithm detects small incremental changes by maintaining cumulative sums of detectable positive or negative shifts from the mean of each data point in the sequence. The threshold for detectable shifts was determined by the product of mshift and the standard deviation of the sequence. Significance was established when the upper or lower cumulative sum of shifts exceeded a threshold based on the product of climit and the standard deviation of the sequence.

**Dopamine changes during learning and extinction.** Measurements of DA release (dLight1.3b $\Delta F/F$, Fig. 7) were performed and analyzed similarly to ACh data. For the analysis of DA signals, 29 fibers in the aDS were selected from a total of 150 fibers within the striatum (out of 324 implanted fibers), based on the criteria AP > 0 mm, ML < 2 mm and DV > -4mm. These fibers were selected from 4 mice (10, 6, 6, 7 fibers in each mouse respectively) implanted with arrays. Other fibers were ignored for this study and anatomical selection criteria was defined only with respect to the regional ACh signaling patterns in the ACh3.0 group and blindly with respect to the DA signals. Dips in DA release were identified as the minimum values of the trial averaged dLight1.3b $\Delta F/F$ within a time window 0.2 to 1.5 s after cue onset or spontaneous unrewarded lick bouts. The latencies of DA dips to spontaneous unrewarded licks bouts in the ITI were calculated as the time elapsed between the ITI lick bout onsets and the minimum $\Delta F/F$, averaged across trials (Fig. 7k). The latencies between the increase in cue-evoked DA dips and decrease in lick indices during extinction were determined using the same analysis as for ACh (see above, Fig. 7f).

**Histology.** At the end of glutamate recordings and TelC behavior experiments, mice were injected intraperitoneally with Euthasol (400 mgkg$^{-1}$, Covertus Euthanasia III) then perfused intracardially with phosphate-buffered saline (PBS 1%, Fisher Scientific), followed by paraformaldehyde (PFA 4% in 1% PBS, Fisher Scientific). The brains were post-fixed overnight in 4% PFA dissolved in PBS and then transferred to a solution of 40% sucrose in 1% PBS (until the brains sank). The brains were sliced (50 μm thickness) with a cryostat (Leica CM3050 S). The coronal sections were then mounted on super frost slides and cover slipped with Vectashield antifade mounting medium (Vector Laboratories, H-1900). TelC (Fig. 6b) and iGluSnFr (Fig. 8b) fluorescence were not immuno-enhanced; confocal images were acquired on a Zeiss LSM 800 laser scanning confocal microscope.

### Reporting summary
Further information on research design is available in the Nature Portfolio Reporting Summary linked to this article.

## Data availability
Source data are provided with this paper and also available on Zenodo (https://doi.org/10.5281/zenodo.14728851). Source data are provided with this paper.

## Code availability
The code used and generated in this study is available from GitHub (https://github.com/craggASAP/Pavlovian_Associations.git) and Zenodo (https://doi.org/10.5281/zenodo.15166878).

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

## Acknowledgements

This work was supported by Aligning Science Across Parkinson's (ASAP, ASAP-020370) through the Michael J. Fox Foundation for Parkinson's Research (MJFF), Stanley Fahn Young Investigator Award (PF-SF-JFA-836662), Klingenstein-Simons's Foundation fellowship, Whitehall Foundation Fellowship, National Institute of Mental Health (R01 MH125835) to M.W.H; NIMH F32MH120894 to M.-A.T.V. We thank the Boston University Centers for Neurophotonics and Systems Neuroscience for financial and technical support, Micro CT core, especially Sydney Holder, for providing equipment and technical expertise for micro-CT scanning and Boston University Animal Science Center for providing central laboratory and animal care and support resources. We thank Glenda Smerin and Anosha Khawaja-Lopez for their technical help. We thank Jason Climer and Daniel Dombeck for the motion correction algorithm code, Lynne Chantranupong and Bernando Sabatini for providing the TelC virus, and Nicolas Tritsch and Eleanor Brown for the helpful discussions of this work.

## Author contributions

Conceptualization - M.W.H., S.B.; Methodology - S.B., M.T.V., M.W.H.; Software - L.Z., M.T.V.; Formal Analysis - L.Z., S.B., M.W.H.; Investigation - S.B., M.T.V., K.T., B.M.G., C.A.N.; Data Curation - S.B., L.Z.; Writing Original Draft - S.B., L.Z., M.T.V., M.W.H.; Visualization - S.B., L.Z., M.W.H.; Supervision - M.W.H.; Funding Acquisition - M.T.V., M.W.H.

## Competing interests

The authors declare no competing interests.
