## [Transparent Peer Review file · Nature Communications]

Distinct spatially organized striatum-wide acetylcholine dynamics for the learning and extinction of Pavlovian associations

Corresponding Author: Dr Mark Howe

Version 0:

Reviewer comments:

Reviewer #1

(Remarks to the Author)

Bouabid S., et al. monitored acetylcholine dynamics across the entire striatal structure evoked by reward, reward-associated sensory cues, and their extinction using an acetylcholine sensor combined with an array fiber photometry system in mice. They found that acetylcholine dynamics differed among striatal regions. In the later experiments of this study, acetylcholine, dopamine, and glutamate dynamics in the anterodorsal striatum (aDS) were tested. Inhibition of acetylcholine release impaired extinction with low-probability rewards.

The techniques and data presented were excellent and convincing. I appreciate the demonstration of the mirrored dopamine and acetylcholine dynamics in the aDS. However, the weakness of this manuscript is that there are no significant conceptual advances. Additionally, the array fiber photometry approach was not fully utilized. Although the authors presented striatal region-specific acetylcholine dynamics, the loss of cholinergic function was tested only in the aDS. Overall, I felt the rationale of this study was not clear.

Major concerns

Regarding the results, definitions of each striatal region are required. From the given figures, it was difficult to identify the specific striatal regions mentioned in the text. Maps showing the distribution of signal components over striatal regions are beautiful; however, it is difficult to discern the distribution of each component due to many color overlaps. For example, the authors claimed that early peaks in the posterior ventral striatum occurred at longer latencies than peaks in other regions (Extended Data Fig. 2b), but I couldn't see this from panel b. Instead, I noticed that the yellowish color points are more prominent in the anteromedial-ventral part.

I wonder if the experimental design of Fig. 3 could demonstrate the prediction error signal encoded by acetylcholine.

TeIC-mediated inhibition of acetylcholine release should be verified. I was surprised that TeIC-mediated inhibition did not affect learning, as the lack of acetylcholine may reduce dopamine release. In the aDS, does inhibition of acetylcholine release change dopamine dynamics?

The reason for examining glutamatergic release in the aDS was not clear. However, the data on tone-evoked acetylcholine release in the aDS are interesting. Since the aDS does not receive excitatory inputs from the auditory cortex, the results using iGluSnFR make sense. I wonder how cholinergic neurons are activated by tone without excitatory input.

Minor comments

- l.150-154: The authors claimed that some locations showed cue selectivity (Extended Data Fig. 3h); however, there is no region information in panel h. The following statement (l.152-153) was also difficult to discern from panel f.
- l. 167-171: The statement seems to refer to overall striatal regions, but Fig. 3e analyzed only the anterior region. When data are obtained from specific regions, the authors should mention this in the text.
- l. 175: "ACh peaks in the posterior ventral striatum increased with learning to the cue." Which data indicate this statement?

- Were the data in Fig. 4d and Extended Data Fig. 6a from only the aDS?
- The given representative traces indicate that the kinetics of ACh dynamics evoked by tone were largely different from those evoked by light (Fig. 2f, Extended Data Fig. 3, Fig. 4f-i, Extended Data Fig. 6c, Extended Data Fig. 8b).
- The distribution of dip counts evoked by light and tone are different.

Reviewer #2

(Remarks to the Author)

The manuscript by Bouabid et al. presents an insightful study on the dynamics of acetylcholine (ACh) sensors across a wide area of the striatum. By employing multi-fiber photometry recording, this work offers a comprehensive dataset that sheds light on the role of ACh signals in learning processes throughout the striatum. The authors effectively report on ACh signals in response to unexpected rewards, cues, and rewards during Pavlovian learning, as well as during extinction. Additionally, they provide valuable insights into the involvement of ACh signals in behavior during the learning process and the dissociation between ACh and glutamate signals before and after learning.

While many findings from this study are consistent with previous electrophysiological recordings, the extensive coverage of recording areas provides new information that has not been previously uncovered due to limitations in the number of recording sites and areas in the striatum. The study is well-designed and well-written, but given the large volume of data, the presentation of the results could benefit from further refinement.

Here are a few points:

1. Please consider enhancing the clarity of data presentation. For example, in Figure 1, it would be helpful to label 'animals with the unpredicted water' either in words or with a small drawing within the figure.
2. It would be beneficial to include example traces of different types of Chl responses (e.g., early peak only, dip only, early peak and dip) alongside the color plots (e.g., Fig 1h, 2h, 3c, 4e, 5e). I understand this might be challenging given the already busy panels, but it could significantly enhance the clarity of your results.
3. For the latency to the early peak (Extended Data, Figure 2), it would be great to use example traces from the anterior or tail of the ventral striatum and align them on the same panel. This would allow readers to make direct comparisons more easily.
4. While it is interesting to see that TelC lesioning of ACh release impairs the extinction of learned behavior, this alone does not imply that the elevation of ACh in the aDS contributes to the extinction of learned Pavlovian responses. Since TelC injection diminishes ACh dynamics, the evidence primarily shows that a lack of ACh release can impair the extinction of learned behavior.
5. The iGluSnFr data is particularly intriguing. Could the authors elaborate on why only aDS iGluSnFr signals were monitored and not other parts of the striatum? Additionally, while the dip in DA may cause disinhibition of ACh release in the striatum, this does not fully explain the changes in ACh sensor signals. Is it possible that synaptic plasticity could lead to different ACh release patterns, even if glutamatergic input remains consistent, as suggested by Zhang et al., 2018 in Neuron?

Reviewer #3

(Remarks to the Author)

Bouabid et al examine striatal acetylcholine dynamics during appetitive Pavlovian conditioning that includes multiple learning stages spanning initial acquisition and updating of probabilistic reward contingencies. They use their recently developed approach with high-density multi-fiber arrays to simultaneously record across many sites throughout the striatum with fiber photometry. This provides an unprecedented spatial coverage throughout this large subcortical brain region, representing one of the most comprehensive data sets to date; fiber photometry and neurotransmitter sensor methods at their best. The authors find evidence for multiphasic ACh responses to cues and rewards that is both consistent with prior literature and reveals novel evidence for intriguing spatiotemporal heterogeneity. Observing particularly prominent 'late peak' responses emerge in the anterior region of dorsal striatum following a downshift in reward probability, they then focus on this subregion for potential links to flexible behavioral updating. Perturbing ACh release in the aDS prevented animals' reduction in conditioned responding following this reward probability downshift, consistent with a role in behavioral flexibility. Additional recordings of dopamine and glutamate sensors in this aDS subregion suggest a preferential influence of DA in potentially driving learning-related changes in ACh.

These experiments provide an impressive amount of data, are well designed and executed. The manuscript presents numerous detailed analyses, and is generally quite thorough, thoughtful, and well-written. My comments and questions below primarily would entail additional analyses of existing data, intended to test related predictions that may help further strengthen the authors' interpretations of this comprehensive data set, and/or questions to expand & deepen the Discussion text. While it would be great to see these new results for my suggestions below, few if any seem absolutely essential, as the existing figures and analyses are already quite extensive. Future work certainly can include causal experiments with greater temporal specificity, additional mechanistic investigations, and more complex behavioral tasks. I would support publication of this manuscript nearly as is with potentially minor revisions, and would welcome responses in consideration of any subset

of the following:

In addition to extensive analysis of cue-evoked responses throughout the multiple task stages, the authors examine responses to reward delivery during initial acquisition, finding reduced ACh dips to rewards as learning progresses (Fig 3). How do these reward-evoked responses following the cues compare to the responses to the 8 non-contingent / unpredicted reward deliveries also interspersed during ITIs in these sessions?

Likewise for the probabilistic sessions' ACh responses at trial outcome: Do the ACh dip amplitudes (or other components) in response to reward delivery and omissions reflect modulation by the cued reward probability / expectation? E.g., larger vs. smaller ACh responses to the same reward delivery following low- vs. high-probability cues; and for reward omissions as a function of these cued probabilities?

Are there any additional noteworthy changes in ACh dynamics following the reversal during the probabilistic phase? The analyses seem to primarily focus on the initial acquisition and transition from deterministic to probabilistic rewards, generally pooling across the 20/80% and 80/20% sessions. Beyond the downshift at that first transition to probabilistic, this reversal now has a big upshift in probability for one cue too. The animals' conditioned licking (Fig 4d) appears to rapidly adapt to this one contingency reversal, considerably faster than the initial acquisition. Do certain components or locations of ACh responses show particularly prominent responses to 'unexpected' rewards following previously low-probability cues in the first few trials after the reversal?

The scatter plots in Fig 3e are a helpful summary of significant dips to cues and/or rewards across sites. Could this be expanded in a supplemental figure to both include the early and late peak components, and maybe also show the non-significant sites in gray (to convey their relative proportion as in Fig 3d, etc). Is there a straightforward way to also generate spatial maps depicting the extent of these often negatively correlated changes for reward vs. cue responses across learning for each of the 3 response components? (Though I realize that may be a few too many dimensions to depict in panels like these)

Are individual differences in behavioral flexibility / learning rate associated with specific features of the ACh signals in certain striatal subregions? I.e., are there any detectable signatures in components of the ACh response dynamics in particular striatal locations that are predictive of which individual animals learn faster or slower during the initial acquisition phase, the downshift to probabilistic contingencies, and/or the reversal?

The one potential additional experiment I'll propose pertains to the one causal experiment in this paper: The authors focus on the aDS, which is well justified given the prominent changes observed across learning in this striatal subregion, particularly the emergence of larger late peaks following the transition to probabilistic reward contingencies. For another striatal subregion where there are still notable ACh responses but less change across learning (e.g. tail of striatum), would a similar TelC perturbation of ACh here have less effect on conditioned responding updates in this same behavioral task? Adding in a comparison group in this behavioral experiment could help strengthen interpretation regarding the importance of the aDS and the specificity of its role in this particular task, even if more extensive examination of ACh function in the tail or other comparison subregions are left for future work.

The Discussion includes a fairly measured mention of DA regulation of ACh and their potential downstream effects on striatal plasticity. Given much recent interest in local striatal ACh-DA interactions and at least some controversy regarding the causal nature of these interactions, how do the authors see their findings situated in the current literature on ACh regulation of DA as well? Several such relevant papers are cited but not discussed in detail in this context. Given that only the aDS DA recordings were analyzed here (29 fibers out of 150 total in the striatum), would the authors expect wider variation in ACh-DA correlations and/or causal interactions across additional striatal subregions beyond the aDS?

Considering the DA (dis)inhibition of ACh release via D2 receptors that the authors do discuss, in combination with heterogeneous patterns of DA release across the striatum arising from diverse DA subtypes with quite distinct responses to salient stimuli & movement-related activity (e.g., Azcorra et al. 2023), to what extent would the authors predict that aspects of the diverse ACh responses observed throughout this paper also may be due in part to these distinct patterns of DA release across striatal subregions?

Is it also worth noting more directly in the text that the current ACh and DA recordings were all in separate animals, so the overlaid traces in Fig 7e & 7k were not recorded simultaneously within the same mice. Nice opportunity for future work with dual-color recording of ACh and DA sensors simultaneously and comprehensively across more striatal subregions to further examine when, where, and under what conditions ACh and DA may be directly interacting, correlated but independently regulated, or temporally uncorrelated.

In these high-density multi-fiber ACh recordings throughout the striatum, is there evidence for spatiotemporal waves, e.g. as reported by Rehani et al (2018) and Matityahu et al. (2023)? Do the direction of any such waves differ between types of behavioral events (cues, rewards, lick bouts, etc)? Or would future work also benefit from comparisons with instrumental tasks in addition to Pavlovian, as for DA waves examined by Hamid et al. (2021)?

Extended Data Figures 4 & 5 provide valuable controls, including analysis of movement-related effects on the ACh signals at cue and reward delivery, as well as potential artifacts also observed with mutant sensor recordings. The potential impact of each indeed appears rather minimal, supporting the robustness of the main results. For the top analyses comparing ACh response amplitudes during large vs. small deceleration, could the authors also provide the spatial maps for the three response components? (The maps in h are all for the mutant null sensor). For the minority of sites where the deceleration size did have some significant effect, is there any notable spatial organization?

And for 'spontaneous' accelerations / decelerations during ITI periods (separate from the unrewarded lick bouts also analyzed later), how do the amplitudes of ACh responses during these movements compare to the cue and/or reward responses? Given the field's interest in ACh and DA signals during spontaneous movement (including prior work from the senior author), in the Discussion it could be informative to briefly comment on the extent to which these salient stimuli vs. movements dominate the ACh signals (and any spatial differences).

Minor:

When describing the behavioral indices of learning (Results paragraph ~lines 115-124), also mention & refer to Extended Data Fig 3 for learning to the tone cue too.

How many sessions on average were included as pre- vs post-learning? The mean (+/- SEM) for the group perhaps could be stated in passing in the Results text. How many intermediate sessions were excluded, where the behavioral lick index for one cue (but not both) was significantly different from the first session? Were such sessions primarily due to different learning rates between cue modalities for a given animal, or differences in session 1 lick index values?

Double-check the wording/grammar in the sentence starting in line 98 ("Only a small fraction of fibers showed a sparse statistically significant changes...") – may be just a singular/plural agreement issue, but something reads slightly off here. Also missing a closing parenthesis?

Add 'Flex' in the virus labels for the TeLC & iGluSnFR AAVs in the Fig 6a & 8a schematics? And consistent nomenclature for the tetanus toxin abbreviation: currently TeCl in Fig 6a, TeC most often throughout the text, and TeLC elsewhere.

The sites targeted for these TeC, DA and glutamate experiments all appear fairly medial. Would it be more precise to label this as 'aDMS' rather than 'aDS' throughout?

The blue colors used for the 'Early peak only' and 'Dip and late peak' locations are a bit difficult to differentiate (Figs 1h, 2h, 3c, 4e, perhaps moreso for the text than for the dots themselves).

The legend for Fig 5e could help better clarify what the left vs right panel columns are (short vs long latency peaks?). Are the dip responses mapped in a figure somewhere for these unrewarded lick bouts? Post-learning responses (aside from the post-vs-pre differences in 5f)?

Methods line 738: "DA<-4mm", DA should be DV

Reviewer #4

(Remarks to the Author)

This manuscript explores the role of acetylcholine (ACh) in the striatum in behavioral flexibility during the learning and extinction of Pavlovian associations. The authors investigate ACh release and its multi-phasic response to both salient and reward-related cues, as well as rewards. Using an impressive novel technique that allows to record from up to 99 micro-fibers simultaneously, the authors sample ACh release across the whole extent of the striatum. The results show region-specific changes during different phases of learning and reward consumption and during extinction. ACh release related to reward delivery was found to be widespread but more restricted during the presentation of cues that signal reward. Notably, ACh release in the anterior dorsal striatum (aDS) shows a late peak during extinction, suggestive of a negative reward prediction error triggered by a decrease in dopamine-mediated inhibition. This study thus reveals significant differences in the dynamics of ACh across striatal regions and contributes to the mechanistic understanding of the role of ACh in behavioral flexibility. The quality of the data is exceptional, the experimental approach is rigorous, and the significance is high. However, the manuscript can be improved by organizing the results differently and clearly summarizing the main findings for each region and condition, as explained below.

Specific comments:

The manuscript is very rich in data, covering a large amount of event related responses in a variety of conditions, making the text difficult to follow. Several variables are presented, which include phases of training, behavioral events, types of cues and striatal regions. The authors use stereotaxic maps to represent changes in ACh across striatal regions but it is difficult to extract the significance of each plot. Simplified figures providing an average of the phase-specific changes in ACh release corresponding to the many different conditions described above would help the reader to get a comprehensive overview of the results and draw their own conclusions and make comparisons.

Related to the point above, the striatal regions sampled for each condition, illustrated by the fiber location, are not consistent across figures. The maps used to illustrate the changes in ACh signaling across striatal regions are repetitive and offer little helpful information. The description in the text is sparse and the patterns of topographical features scarcely described, making the text loosely related to the maps. Using the maps as supplementary figures and instead illustrate the changes in ACh release across a limited number of striatal regions (e.g., aDS, ventral striatum, tail of the striatum, etc.) would make the data clearer and the significance more evident.

Regarding the calculation of dF/F (lines 87-89 and methods), by using such a low percentile (the 8th percentile) as the baseline, the authors are making the signal detection highly sensitive to small variations in the fluorescence signal. This means that even minor increases from this low baseline can be interpreted as significant changes, which might be just noise or minor fluctuations. This increased sensitivity raises the risk of misinterpreting noise or artifacts as genuine signals, especially if the baseline is unstable or exhibits significant fluctuations. While this approach may be justified if the goal is to detect subtle changes in fluorescence, it introduces the concern that these subtle changes could be mistaken for motion artifacts. The authors are addressing motion artifacts using a null-sensor from an external control group rather than an internal control. Given this setup, how can the authors be confident that the motion artifacts in the external control group are sufficiently similar to those in the experimental group? This is crucial for ensuring that subtle changes detected using the 8th percentile baseline are truly reflective of biological signals rather than artifacts.

Data for reversal learning is shown in Fig. 4 (panels a, b and c), but there is no description of the results in the text and no further mention in the manuscript.

The correlation between the dopamine dip and the ACh late peak is in agreement with current literature, but this correlation seems to occur only in aDS. However, most dopamine neurons have been shown to decrease their firing during negative RPE thus releasing cholinergic interneurons from inhibition across most striatal regions. Why wouldn't CINs in other striatal regions show an increase in the amplitude of the late peak, as they do in aDS?

Lines 193-195: The authors state that "these correlations could not account for the patterns of cue and reward evoked release that we observed", by showing that "the magnitude of the cue-evoked dF/F does not vary with deceleration magnitude" (Ext Data Fig 4). This does not rule out that ACh release might be sensitive to, or signaling locomotion, e.g. deceleration. It only shows that it does not distinguish between slow and fast deceleration. The authors should provide data showing ACh release at locomotor events (e.g., deceleration) during the ITI or spontaneous movements are not coinciding with cue onset.

Lines 690: Was the statistical analysis conducted by adding all fibers corresponding to a region out of the 295 fibers that were analyzed, or were signals from fibers corresponding to a region for each animal first averaged and then averaged across animals? Please provide a table that indicates the number of fibers for each animal for each region.

Line 312: "...these results indicate that DA signals in the aDS exhibit emerging dips to cues after partial extinction and spontaneous ITI licks prior to learning which precede ACh elevations in the same striatal region". Since these measurements come from different cohorts and related to the median 27 trials prior to lick change, this temporal correlation needs to be expressed more cautiously or shown within the same animal.

Lines 70-71 and 449: The text describes that the 55-99 optical micro-fibers (50um diameter) are distributed uniformly, collecting about 100um axially and 25um radially from each tip without overlap. The authors refer to Figure 1a,c, which does not really show how this volume is calculated and the methods solely provides references. Since this is crucial to the experiment, please provide a brief explanation how the volume was calculated.

Line 166: The authors state that "Changes to the initial and late peaks [at reward delivery] were more variable and less widespread" but write in line 173-175: "changes in the reward response with learning were slightly more widespread than for the cues, indicating that learning related changes to cues and rewards are partially decoupled and vary across regions". This sounds contradictory. Please clarify.

Line 187 does not refer to figure 2.

Line 211: "Elevations in late peaks following partial extinction were present for at least one fiber in 7 out of 8 mice for the light and 5 out of 8 mice for the tone (26/53, 49% and 17/53, 32%...)". How relevant is the signal measured from 1 fiber out of a minimum of 55 fibers and how do the authors get to the total numbers of 26 and 17? Please clarify.

Extended data Fig 3h: The legend states that colors indicate significance but panel says something else.

Line 376: "we found that persistent inhibition of ACh release from aDS cCINs prevented the down-regulations of licking during partial extinction (figure 6)". How can the authors here infer causality?

The map in Fig. 2h suggest a pattern where dip only neurons are spatially clustered in the pre-learning phase, but this is not mentioned in the manuscript. If the purpose of presenting maps is to illustrate distribution patterns, quantitative analysis should be used.

Fig. 6: Why are TelC animals different to controls in the percent time liking during the pre-learning phase. How is this impacting the results during post-learning and extinction?

Reviewer #5

(Remarks to the Author)

Reviewer #6

(Remarks to the Author)

Version 1:

Reviewer comments:

Reviewer #1

(Remarks to the Author)

The authors have addressed all the concerns I raised. Both the manuscript and data presentation have been significantly improved.

Reviewer #2

(Remarks to the Author)

The authors have adequately addressed all my concerns. Congratulations!

Reviewer #3

(Remarks to the Author)

The authors have been very responsive to the reviews. They have improved the data presentation and included additional analyses & discussion. I congratulate them for this comprehensive data set, and I look forward to seeing more future directions with these exciting approaches.

Reviewer #4

(Remarks to the Author)

The substantial reorganization and major revisions have significantly improved the clarity and impact of the manuscript, allowing the excellent data to be fully appreciated. Initially, the key conceptual advances were somewhat obscured by the complexity of techniques and the sheer volume of data. However, in its revised form, the manuscript presents a compelling and well-structured narrative that effectively highlights the significance of the findings.

We particularly appreciate the more accessible presentation of the distribution of ACh signals across the striatum in the main text, which now directs focus to the most relevant areas while acknowledging the value of keeping the detailed plots in the supplementary materials for completeness. The dataset remains highly impressive, and the analyses are both extensive and rigorous, with well-executed controls. Despite the large amount of data and intricate details, the figures are now much more digestible and contribute effectively to the manuscript's readability.

All our comments and concerns have been thoroughly addressed, either in the rebuttal and/or with revisions in the manuscript, and we strongly support the publication of this manuscript in its current form.

Reviewer #5

(Remarks to the Author)

Reviewer #6

(Remarks to the Author)

Response to Reviewers

We thank the reviewers for their positive feedback and valuable suggestions, which have greatly expanded and clarified our main conclusions and improved the manuscript. In the following sections, we detail the additional experiments and analyses conducted, along with our point-by-point responses (in blue) to each of the reviewers's comments (in black). Major revisions were done on all sections of the manuscript, and sections pertaining to specific reviewer concerns are cited in the below responses.

REVIEWER COMMENTS

Reviewer #1 (Remarks to the Author):

Bouabid S., et al. monitored acetylcholine dynamics across the entire striatal structure evoked by reward, reward-associated sensory cues, and their extinction using an acetylcholine sensor combined with an array fiber photometry system in mice. They found that acetylcholine dynamics differed among striatal regions. In the later experiments of this study, acetylcholine, dopamine, and glutamate dynamics in the anterodorsal striatum (aDS) were tested. Inhibition of acetylcholine release impaired extinction with low-probability rewards.

The techniques and data presented were excellent and convincing. I appreciate the demonstration of the mirrored dopamine and acetylcholine dynamics in the aDS. However, the weakness of this manuscript is that there are no significant conceptual advances. Additionally, the array fiber photometry approach was not fully utilized. Although the authors presented striatal region-specific acetylcholine dynamics, the loss of cholinergic function was tested only in the aDS. Overall, I felt the rationale of this study was not clear.

Author response (1.1). We understand that the conceptual advances provided by our study may not have been clearly enough communicated, so we substantially re-organized the figures, added new analyses, and made major revisions to the text. These additions have further clarified our primary conclusions and have led to important new insights. The main objective of our study is to establish *where and how* naturally occurring ACh dynamics are expressed across functionally distinct striatal regions to support associative Pavlovian learning and extinction. We provide the largest scale and most comprehensive survey of striatum-wide ACh dynamics to date and identify changes in ACh signaling in different signal phases and task events across distinct anatomical gradients. New analyses, suggested by this reviewer and others, have prompted a refinement of our primary claims and a more precise quantification of the spatial organization of signal changes. Most notably, we determined that signals consistent with positive and negative RPEs to cues and rewards during Pavlovian learning and extinction are expressed within distinct components of the ACh signal and are localized primarily to a specific region of the aDS. We identified novel cue and unrewarded lick evoked ACh increases in the aDS, consistent with negative RPE encoding, and provide evidence for a mechanism for their generation (through coincident glutamate input and DA dips). Together, these findings provide critical new insights into how ACh signaling may contribute to enabling associative learning (through dips) and promoting new plasticity (or erasing existing changes) to downregulate learned cue-reward associations when they are no longer valid.

Major concerns

Regarding the results, definitions of each striatal region are required. From the given figures, it was difficult to identify the specific striatal regions mentioned in the text. Maps showing the distribution of signal components over striatal regions are beautiful; however, it is difficult to discern the distribution of each component due to many color overlaps. For example, the authors claimed that early peaks in the posterior ventral striatum occurred at longer latencies than peaks in other regions (Extended Data Fig. 2b), but I couldn't see this from panel b. Instead, I noticed that the yellowish color points are more prominent in the anteromedial-ventral part.

Author response (1.2). We agree with the reviewer that our reference to striatal 'regions' in the text were not precisely defined by specific anatomical coordinates. Indeed, we wanted to avoid defining striatal regions based on arbitrary anatomical boundaries but instead show the distribution of signals across the striatum volume in an unbiased way. However, we recognize that showing every fiber in the colormaps made it somewhat difficult to identify, quantify, and reference regional patterns in the text. To address this, we added several new analyses and made major changes to the text. First, to better capture and visualize the distribution of signal changes during learning and extinction, we replaced the maps in the main figures showing changes for each individual fiber with maps showing a smoothed weighted mean of changes collapsed across the axial and sagittal planes (e.g. Fig. 2j,k). From these maps, we could define regions over which the strongest concentration of changes occurred for each signal component. Detailed spatial data for individual fibers was moved to the supplement. We feel that these more detailed plots should remain in the paper for transparency and to highlight the nuances of the spatial distribution of changes to inform future studies. Second, to explicitly quantify and compare the variation of signal changes across all three striatal axes, we used a generalized linear model approach (see Methods, e.g. Fig 2l,m). This provided explicit statistical support for our claims that changes differed across specific spatial gradients for particular signal components and cue modalities during learning and extinction.

I wonder if the experimental design of Fig. 3 could demonstrate the prediction error signal encoded by acetylcholine.

Author response (1.3). We thank the reviewer for raising this important point and have added additional analyses and figure panels to more explicitly test for reward prediction error (RPE) signaling. This analysis has led to new insights and clarified our main conclusions. To summarize, we conclude that signal changes at cues, rewards, and unrewarded ITI licks consistent with RPE encoding are predominantly localized to the anterior dorsal striatum (aDS), particularly for the light cue (Figs 3d, 5g, S3e, and S5f). Signal changes for positive and negative RPEs are primarily expressed in the dip and the late peak components of the aDS ACh signals respectively. These signal changes are opposite in polarity to RPEs expressed in dopamine neuron firing and release. The early peak component of the ACh signal (strongest in the ventral striatum), in contrast, increased to the cues during learning and decreased during extinction (in the same direction as DA in previous studies) but did not change consistently at reward. Thus, early peak changes were only partially consistent with positive RPE signaling.

Moreover, our new analyses identified modality selectivity in positive and negative RPE signaling at some locations in the dorsal striatum (e.g. Fig. S4e, S6f). These results have important implications for how ACh may be influenced by DA and shape plasticity within spatially restricted regions, perhaps fine-tuning the impact of a more spatially homogenous DA RPE signal (lines 483-492). We have revised all sections of the discussion to further expand on the consequences and potential mechanisms for RPE encoding within distinct ACh signal components and regions.

TelC-mediated inhibition of acetylcholine release should be verified. I was surprised that TelC-mediated inhibition did not affect learning, as the lack of acetylcholine may reduce dopamine release. In the aDS, does inhibition of acetylcholine release change dopamine dynamics?

Author response (1.4). TelC mediated inhibition of striatal cholinergic interneurons and ACh release with the same viral construct has been validated in a previous study (Chantranupong et al. 2023). However, we conducted an additional validation experiment in two mice to demonstrate that TelC expression in the aDS produces a loss of ACh peaks and dips to unpredicted reward delivery (Fig 6c). Our validation confirms a general elimination of aDS ACh release dynamics. We agree that it would be interesting to know the impact of ACh inhibition on DA release, as previous studies have provided conflicting evidence on this point, but this is beyond the scope of our current study. Learning predominantly resulted in changes in ACh dips to cues and rewards in the aDS, which we (and others) hypothesize may be controlling spatiotemporal windows for DA driven plasticity to occur. If this hypothesis is correct, general suppression of ACh signaling would not necessarily be expected to negatively impact initial associative learning.

The reason for examining glutamatergic release in the aDS was not clear. However, the data on tone-evoked acetylcholine release in the aDS are interesting. Since the aDS does not receive excitatory inputs from the auditory cortex, the results using iGluSnFR make sense. I wonder how cholinergic neurons are activated by tone without excitatory input.

Author response (1.5). The rationale for examining glutamate release onto CINs in the aDS was to determine whether the elevations in late peak ACh release magnitudes during extinction could be driven by increases in excitatory drive from the late learning to extinction phases. We found that glutamate release onto aDS CINs for light cues displays early and late peaks, but the late peak does not change during extinction (Fig. 8). These findings indicate that changes in excitatory drive from upstream inputs could not explain the elevations in aDS ACh release during extinction, but that the late peak of excitation may be 'unmasked', perhaps by the emergence of DA dips. Overall, the iGluSnFR measurements provide insight into the mechanism by which ACh release dynamics are controlled to promote the downshifting of learned cue values.

As the reviewer correctly points out, there is little detectable elevation of glutamate release onto the aDS CINs to the tone cue, consistent with the lack of strong direct inputs from auditory

cortex to the aDS. This result is also consistent with the relatively weaker early and late ACh peaks to the tone in this region (Figs S2d,f,g and S4b,c,e). Some locations in the anterior striatum (particularly ventral) did show ACh increases to the tone cue across learning and extinction, which may be driven by disinhibition through other inputs or by glutamate release from non-auditory cortical regions (which was not detected by our iGluSnFR approach). We also note that the strongest late peak increases to the tone during extinction were concentrated in a partially distinct region from the increases to the light cue, perhaps because of differences in glutamatergic input to the two regions (Figs. 4h,i and S4c,e). We plan to further explore how regional differences in glutamate release relate to striatum-wide patterns of ACh release in future studies.

Minor comments

- l.150-154: The authors claimed that some locations showed cue selectivity (Extended Data Fig. 3h); however, there is no region information in panel h. The following statement (l.152-153) was also difficult to discern from panel f.

Author response (1.6). We agree with the reviewer that regional differences in cue selectivity should be more explicitly illustrated and quantified. In response, we conducted new analyses that better highlight the modality specificity across individual locations and striatal axes (Figs. 2m, 4l, S2g, and S4e, Response 1.2).

- l. 167-171: The statement seems to refer to overall striatal regions, but Fig. 3e analyzed only the anterior region. When data are obtained from specific regions, the authors should mention this in the text.

Author response (1.7). We agree that the relationship between cue and reward changes should be more explicitly assessed at individual locations. We have removed these figures and revised the statements regarding putative RPE encoding by changes in ACh release, as described above in Response 1.3.

- l. 175: "ACh peaks in the posterior ventral striatum increased with learning to the cue." Which data indicate this statement?

Author response (1.8). This statement refers to the increases in the early ACh peak concentrated in a region of the ventral striatum (e.g. Fig 2j). We agree that the use of 'posterior' is confusing here and have eliminated this term when describing the increases in the peak. We meant to contrast with a relatively more anterior region of the ventral striatum which appears more similar to anterior dorsal (Fig S1g). We have now provided more explicit quantification of the variation in signaling changes across different anatomical gradients (see Response 1.2) and have updated the descriptions in the text accordingly.

- Were the data in Fig. 4d and Extended Data Fig. 6a from only the aDS?

Author response (1.9). No, these data were collapsed across all fibers. None of the quantification of ACh signaling was done only on aDS fibers.

- The given representative traces indicate that the kinetics of ACh dynamics evoked by tone were largely different from those evoked by light (Fig. 2f, Extended Data Fig. 3, Fig. 4f-i, Extended Data Fig. 6c, Extended Data Fig. 8b).

Author response (1.10). To address this, we added new quantification of the mean latencies to peaks/troughs for light and tone cues for each signal component during the learning phase (Fig. S1a,b, lines 1479-1490). This analysis revealed that latencies to early peaks and dips (troughs) were indeed significantly different between light and tone cue release. Tone cue responses were slightly slower than light cue on average.

- The distribution of dip counts evoked by light and tone are different.

Author response (1.11). We agree - during initial learning, there were more dips and larger changes in dip magnitudes with learning for the light cue than for the tone. Percentages of fibers are now more explicitly quantified, and we have acknowledged this observation in the text (Fig. 2g,i; Figs. S1a-d and S2c, lines 170-177, 198-200).

Reviewer #2 (Remarks to the Author):

The manuscript by Bouabid et al. presents an insightful study on the dynamics of acetylcholine (ACh) sensors across a wide area of the striatum. By employing multi-fiber photometry recording, this work offers a comprehensive dataset that sheds light on the role of ACh signals in learning processes throughout the striatum. The authors effectively report on ACh signals in response to unexpected rewards, cues, and rewards during Pavlovian learning, as well as during extinction. Additionally, they provide valuable insights into the involvement of ACh signals in behavior during the learning process and the dissociation between ACh and glutamate signals before and after learning.

While many findings from this study are consistent with previous electrophysiological recordings, the extensive coverage of recording areas provides new information that has not been previously uncovered due to limitations in the number of recording sites and areas in the striatum. The study is well-designed and well-written, but given the large volume of data, the presentation of the results could benefit from further refinement.

Here are a few points:

1. Please consider enhancing the clarity of data presentation. For example, in Figure 1, it would be helpful to label 'animals with the unpredicted water' either in words or with a small drawing within the figure.

Author response (2.1). We thank the reviewer for this suggestion and have made improvements to all of the figures to enhance the clarity of the data presentation and to better

highlight and quantify the region specific changes we have observed. We have also made the suggested change to Fig. 1.

2. It would be beneficial to include example traces of different types of CIN responses (e.g., early peak only, dip only, early peak and dip) alongside the color plots (e.g., Fig 1h, 2h, 3c, 4e, 5e). I understand this might be challenging given the already busy panels, but it could significantly enhance the clarity of your results.

Author response (2.2). We thank the reviewer for this suggestion. We included the component presence maps to show details on how different ACh emerged or disappeared over conditions and to highlight that combinations of significant components were present at some locations. However, these classifications are not absolute and do not reflect distinct 'categories' of response types, as the relative magnitudes of each component can vary partially independently across fibers. As we show in the quantification of the magnitude of changes over learning, the components vary across distinct, but partially overlapping 3-dimensional gradients. We agree that triggered averages for specific fibers illustrating specific response profiles and changes with learning are helpful and have included pertinent examples.

3. For the latency to the early peak (Extended Data, Figure 2), it would be great to use example traces from the anterior or tail of the ventral striatum and align them on the same panel. This would allow readers to make direct comparisons more easily.

Author response (2.3). We thank the reviewer for this suggestion. We added this panel to Fig S1e.

4. While it is interesting to see that TelC lesioning of ACh release impairs the extinction of learned behavior, this alone does not imply that the elevation of ACh in the aDS contributes to the extinction of learned Pavlovian responses. Since TelC injection diminishes ACh dynamics, the evidence primarily shows that a lack of ACh release can impair the extinction of learned behavior.

Author response (2.4). We agree with the reviewer and modified the text (lines 374-376) to reflect that the lack of ACh release impairs the extinction of learned Pavlovian responses following a downshift in cue associated reward probabilities.

5. The iGluSnFr data is particularly intriguing. Could the authors elaborate on why only aDS iGluSnFr signals were monitored and not other parts of the striatum? Additionally, while the dip in DA may cause disinhibition of ACh release in the striatum, this does not fully explain the changes in ACh sensor signals. Is it possible that synaptic plasticity could lead to different ACh release patterns, even if glutamatergic input remains consistent, as suggested by Zhang et al., 2018 in Neuron?

Author response (2.5). We agree that monitoring glutamate release onto CINs across the entire striatum could be very interesting and perhaps illuminate which signals and spatial

patterns are locally generated vs input driven. We focused the iGluSnFr measurements only on the aDS in this study, in part because we were concerned that glutamate signals onto CINs would be more difficult to detect with our small diameter fibers given the sparseness of CINs, spatial heterogeneities in glutamatergic input, and the relatively lower SNR of iGluSnFr. We are excited to further explore this question in future studies. We agree that synaptic plasticity is another potential explanation for why the aDS ACh peaks emerged during extinction but the glutamatergic input did not change. We have added additional text to the discussion to point this out (lines 510-514).

Reviewer #3 (Remarks to the Author):

Bouabid et al examine striatal acetylcholine dynamics during appetitive Pavlovian conditioning that includes multiple learning stages spanning initial acquisition and updating of probabilistic reward contingencies. They use their recently developed approach with high-density multi-fiber arrays to simultaneously record across many sites throughout the striatum with fiber photometry. This provides an unprecedented spatial coverage throughout this large subcortical brain region, representing one of the most comprehensive data sets to date; fiber photometry and neurotransmitter sensor methods at their best. The authors find evidence for multiphasic ACh responses to cues and rewards that is both consistent with prior literature and reveals novel evidence for intriguing spatiotemporal heterogeneity. Observing particularly prominent 'late peak' responses emerge in the anterior region of dorsal striatum following a downshift in reward probability, they then focus on this subregion for potential links to flexible behavioral updating. Perturbing ACh release in the aDS prevented animals' reduction in conditioned responding following this reward probability downshift, consistent with a role in behavioral flexibility. Additional recordings of dopamine and glutamate sensors in this aDS subregion suggest a preferential influence of DA in potentially driving learning-related changes in ACh.

These experiments provide an impressive amount of data, are well designed and executed. The manuscript presents numerous detailed analyses, and is generally quite thorough, thoughtful, and well-written. My comments and questions below primarily would entail additional analyses of existing data, intended to test related predictions that may help further strengthen the authors' interpretations of this comprehensive data set, and/or questions to expand & deepen the Discussion text. While it would be great to see these new results for my suggestions below, few if any seem absolutely essential, as the existing figures and analyses are already quite extensive. Future work certainly can include causal experiments with greater temporal specificity, additional mechanistic investigations, and more complex behavioral tasks. I would support publication of this manuscript nearly as is with potentially minor revisions, and would welcome responses in consideration of any subset of the following:

In addition to extensive analysis of cue-evoked responses throughout the multiple task stages, the authors examine responses to reward delivery during initial acquisition, finding reduced ACh dips to rewards as learning progresses (Fig 3). How do these reward-evoked responses

following the cues compare to the responses to the 8 non-contingent / unpredicted reward deliveries also interspersed during ITIs in these sessions?

Author response (3.1). We thank the reviewer for these insightful questions. We conducted additional analyses to directly compare ACh responses to non-contingent/unpredicted reward deliveries with responses to predicted rewards following learning (Fig. S3c,d). Similar to the pre and post learning comparisons, we found that ACh dips to unpredicted rewards were larger compared to predicted reward dips for the majority of fibers (Fig. S3c). We also found that dips were larger following the low probability cue than the high probability cue (Fig. S3d). These results are generally consistent with RPE encoding at reward.

Likewise for the probabilistic sessions' ACh responses at trial outcome: Do the ACh dip amplitudes (or other components) in response to reward delivery and omissions reflect modulation by the cued reward probability / expectation? E.g., larger vs. smaller ACh responses to the same reward delivery following low- vs. high-probability cues; and for reward omissions as a function of these cued probabilities?

Author response (3.2). Unfortunately, analyzing ACh responses to reward omissions during extinction presented a challenge due to alignment difficulties. The variability in licking and velocity changes during omission trials indicated that mice did not consistently estimate the precise time that omitted reward was supposed to be delivered (i.e. they continued to lick with reward anticipation for variable intervals past the time of predicted reward). Additionally, the mice begin anticipatory licking well before the time of expected reward delivery. Thus, accurate alignment of ACh signals to unexpected reward omissions was imprecise, so we could not draw definitive conclusions about ACh responses amplitudes for omissions. However, we did attempt this analysis and observed larger overall ACh release to unpredicted omissions than predicted, but this was somewhat variable across mice and locations and not precisely time-locked to any discrete event, likely for the reasons above, and therefore we did not include this in the manuscript. As an alternative to investigating reward omissions in the task, we analyzed dynamics to spontaneous, unrewarded spout licks in the ITI, which are associated with different expectations of reward from the early to late phase of learning. This analysis has shown larger ACh late peaks in the aDS for ITI lick bout onsets early in learning, relative to late, consistent with a negative reward prediction error (Figs. 5 and S6). See response 3.1 regarding the high and low probability reward.

Are there any additional noteworthy changes in ACh dynamics following the reversal during the probabilistic phase? The analyses seem to primarily focus on the initial acquisition and transition from deterministic to probabilistic rewards, generally pooling across the 20/80% and 80/20% sessions. Beyond the downshift at that first transition to probabilistic, this reversal now has a big upshift in probability for one cue too. The animals' conditioned licking (Fig 4d) appears to rapidly adapt to this one contingency reversal, considerably faster than the initial acquisition. Do certain components or locations of ACh responses show particularly prominent responses to 'unexpected' rewards following previously low-probability cues in the first few trials after the reversal?

Author response (3.3). We agree that reversal learning could produce different response dynamics than the initial extinction phase. However, due to the variability in learning rates across mice and the small number of trials from the time of the probability reversal to the anticipatory licking changes, we could not make definitive comparisons between the time course or spatial patterns of ACh release changes between the initial extinction and probability reversal phases. We did find, however, that the late peak differences for high and low probability cues were qualitatively similar in both the initial extinction and reversal phases, justifying our combining data from reversal and initial extinction phases for all analyses (see example below).

The scatter plots in Fig 3e are a helpful summary of significant dips to cues and/or rewards across sites. Could this be expanded in a supplemental figure to both include the early and late peak components, and maybe also show the non-significant sites in gray (to convey their relative proportion as in Fig 3d, etc). Is there a straightforward way to also generate spatial maps depicting the extent of these often negatively correlated changes for reward vs. cue responses across learning for each of the 3 response components? (Though I realize that may be a few too many dimensions to depict in panels like these).

Author response (3.4). We thank the reviewer for this suggestion, which has led to important new insights and clarified our main claims. We have added new maps to show locations with opposite changes over learning for cues, rewards, and unrewarded ITI licks, indicating putative RPE encoding, for each component (Figs 3d,5g,S3e, and S6f). This analysis has indicated that positive and negative RPEs are encoded in ACh dips and late peaks respectively for a limited subset of fibers localized to the anterior dorsal striatum for light cue associations.

Are individual differences in behavioral flexibility / learning rate associated with specific features of the ACh signals in certain striatal subregions? I.e., are there any detectable signatures in components of the ACh response dynamics in particular striatal locations that are predictive of which individual animals learn faster or slower during the initial acquisition phase, the downshift to probabilistic contingencies, and/or the reversal?

Author response (3.5). This is an interesting question, but we do not have a large enough cohort of mice to make definitive conclusions regarding the relationship of region specific ACh signals to individual behavioral variability in learning or extinction rates. Now that we have pinpointed the regions over which the relevant changes in ACh dynamics occur, we can perform, in future studies, more spatially targeted, higher throughput measurements and manipulations across larger mouse cohorts to address this question.

The one potential additional experiment I'll propose pertains to the one causal experiment in this paper: The authors focus on the aDS, which is well justified given the prominent changes observed across learning in this striatal subregion, particularly the emergence of larger late peaks following the transition to probabilistic reward contingencies. For another striatal subregion where there are still notable ACh responses but less change across learning (e.g. tail of striatum), would a similar TelC perturbation of ACh here have less effect on conditioned responding updates in this same behavioral task? Adding in a comparison group in this behavioral experiment could help strengthen interpretation regarding the importance of the aDS and the specificity of its role in this particular task, even if more extensive examination of ACh function in the tail or other comparison subregions are left for future work.

Author response (3.6). We appreciate the insightful suggestion and agree that comparing the effects of CIN inhibition in different subregions (e.g. ventral or posterior striatum) would strengthen our claims. However, our study is already quite comprehensive, and we feel that our primary conclusions would not significantly hinge on the outcome of additional manipulation experiments. We are planning follow up studies to more comprehensively probe the specific contributions of ACh signals across different striatal regions to behavior and learning.

The Discussion includes a fairly measured mention of DA regulation of ACh and their potential downstream effects on striatal plasticity. Given much recent interest in local striatal ACh-DA interactions and at least some controversy regarding the causal nature of these interactions, how do the authors see their findings situated in the current literature on ACh regulation of DA as well? Several such relevant papers are cited but not discussed in detail in this context. Given that only the aDS DA recordings were analyzed here (29 fibers out of 150 total in the striatum), would the authors expect wider variation in ACh-DA correlations and/or causal interactions across additional striatal subregions beyond the aDS?

Author response (3.7). This is an excellent point and a focus of ongoing work in our lab. We have added some additional discussion on how the regional differences in ACh signals we report could be shaped by DA signals (lines 473-481, 503-514, 516-531). Overall, our results suggest a nuanced interplay in which DA and ACh relationships vary depending on striatal region, local circuitry and inputs in specific behavior contexts. For example, we propose that the late ACh peaks in aDS result from a disinhibition driven by DA dips encoding putative negative RPEs. The emergence of the late ACh peak may also depend on appropriately timed glutamatergic input, which may vary topographically across the striatum (supported by the different late peak topography and glutamate release dynamics for light and tone). Our results

suggest that DA/ACh interactions in the aDS may be antagonistic, with DA inhibiting ACh. This relationship may contribute to the selective encoding of RPEs in the dip and late peak components in this region. However, in the ventral striatum, ACh release increased to cues during learning and decreased during extinction, presumably in the same direction as DA. Thus, DA and ACh interactions may be cooperative in this region (or reflect common excitatory input). Our future work will focus on directly defining natural coordination of DA and ACh across behaviors, with simultaneous recordings, to determine where and when DA/ACh interactions may manifest.

Considering the DA (dis)inhibition of ACh release via D2 receptors that the authors do discuss, in combination with heterogeneous patterns of DA release across the striatum arising from diverse DA subtypes with quite distinct responses to salient stimuli & movement-related activity (e.g., Azcorra et al. 2023), to what extent would the authors predict that aspects of the diverse ACh responses observed throughout this paper also may be due in part to these distinct patterns of DA release across striatal subregions?

Author response (3.8). It is possible that some of the spatial heterogeneity in ACh release we report is driven by heterogeneities in DA release, perhaps originating from different midbrain DA populations, as reported by Azcorra et al and others. However, we believe this is unlikely to explain all of the regional differences we observed, particularly given that previous studies have found weak or subtle influence of DA elevations on ACh and vice versa *in-vivo*. As noted above, however, we hypothesize that DA changes do contribute to changes in specific ACh signal components under particular conditions. Most notably, our evidence suggests that dips in DA to cues during extinction may promote the emergence of ACh late peaks in the aDS which ultimately contribute to plasticity that promotes the downregulation of responding. Conversely, changes in DA increases over learning to cues and rewards may modulate inhibition of CINs and the size of the dips, perhaps contributing to the region selective positive RPE encoding we observed in the aDS. However, this raises the question of why, if positive and negative RPEs are broadly encoded in DA release, these changes in ACh late peaks and dips were not more widely present across the striatum. As the reviewer points out, one reason may be that DA RPEs are also more regionally specific than currently appreciated. However, other factors are also possible (see Author response 4.4 for more discussion of this point). Additional text has been added to the discussion to address these points (lines 473-481,516-520). Future simultaneous DA/ACh recordings and manipulations will be needed to clarify the direct interactions between DA and ACh across striatal regions.

Is it also worth noting more directly in the text that the current ACh and DA recordings were all in separate animals, so the overlaid traces in Fig 7e & 7k were not recorded simultaneously within the same mice. Nice opportunity for future work with dual-color recording of ACh and DA sensors simultaneously and comprehensively across more striatal subregions to further examine when, where, and under what conditions ACh and DA may be directly interacting, correlated but independently regulated, or temporally uncorrelated.

Author response (3.9). We agree and have added additional clarification to the text (lines 392-393, 396-397, 506-508).

In these high-density multi-fiber ACh recordings throughout the striatum, is there evidence for spatiotemporal waves, e.g. as reported by Rehani et al (2018) and Matityahu et al. (2023)? Do the direction of any such waves differ between types of behavioral events (cues, rewards, lick bouts, etc)? Or would future work also benefit from comparisons with instrumental tasks in addition to Pavlovian, as for DA waves examined by Hamid et al. (2021)?

Author response (3.10). We did not explicitly test for waves in ACh release on single trials at cue presentation or reward. However, we report different signal latencies at the cue (consistent across trials) across regions which would result in wave-like dynamics. For example, we found that early peak release components had different latencies across the anterior dorsal-ventral axis, with longer latency peaks in the ventral striatum relative to the dorsal (Fig S1e-g). While this may result from a dorsal to ventral ‘wave’ of release, other explanations are possible. Relative latency differences were not clearly apparent for the onsets of ACh peaks (Fig S1e), meaning that the shorter latency peaks in the dorsal striatum may be the result of truncation of the peak by the longer latency dips, which were larger and more frequent in the dorsal than the ventral striatum (Fig. 2). A non-mutually exclusive possibility is that peaks in the dorsal and ventral striatum are generated by glutamatergic inputs originating from different regions and occurring at different latencies. We looked for, but did not observe, consistent differences across striatal axes in the trough latencies for dips or peak latencies for late peaks at cues or rewards. In sum, while we agree that investigating ACh (and DA) waves at different task periods could be an interesting direction, further analysis (e.g. trial by trial) and experiments, beyond the scope of this study, would be required to properly interpret the temporal dynamics with respect to different underlying mechanisms, functional roles, and task contexts.

Extended Data Figures 4 & 5 provide valuable controls, including analysis of movement-related effects on the ACh signals at cue and reward delivery, as well as potential artifacts also observed with mutant sensor recordings. The potential impact of each indeed appears rather minimal, supporting the robustness of the main results. For the top analyses comparing ACh response amplitudes during large vs. small deceleration, could the authors also provide the spatial maps for the three response components? (The maps in h are all for the mutant null sensor). For the minority of sites where the deceleration size did have some significant effect, is there any notable spatial organization?

And for ‘spontaneous’ accelerations / decelerations during ITI periods (separate from the unrewarded lick bouts also analyzed later), how do the amplitudes of ACh responses during these movements compare to the cue and/or reward responses? Given the field’s interest in ACh and DA signals during spontaneous movement (including prior work from the senior author), in the Discussion it could be informative to briefly comment on the extent to which these salient stimuli vs. movements dominate the ACh signals (and any spatial differences).

Author response (3.11). We agree that exploring movement related ACh release is an important topic - for the purposes of this paper, we focused only on accounting for potential movement contributions to the task dynamics across learning. A proper characterization of movement related ACh release will require deeper analysis and experiments better suited for a separate paper. That said, at the reviewer's suggestion, we conducted additional analysis on ITI periods to address the possibility that changes in locomotor decelerations occurring at the cue or reward could account for some of the learning related changes we observed (Fig. S7). Velocity changes at the cue and reward changed significantly (for some comparisons) from pre to post learning but did not differ from post-learning to extinction or between the two extinction cues (Fig. S7). Therefore, it is possible that dips or peaks related to changing decelerations could be accounting for the apparent changes with learning at cues and rewards (but not during extinction). To test for this, we aligned ACh signals on spontaneous ITI decelerations (without reward delivery or licking) of comparable magnitude to the task. We did observe multi-phasic peaks and dips (minima and maxima) at some locations (Fig. S7d,e). However, significant signal changes were present at a smaller fraction of locations relative to the changes at the cue and reward during learning (Fig S7e). Responses were scattered primarily across the dorsal central part of the striatum (Fig S7d), and the spatial organization did not match any of the task related changes we report with learning. Moreover, the magnitude (min or max) of the signal changes for spontaneous decelerations was much lower at most sites relative to the cue and reward related signal changes we observed (Fig S7e-g). A few fibers in the aDS showed small deceleration peaks that were close in magnitude to the small increases in cue peaks with learning (Fig. S7d,f). However, our main finding was that peak increases were strongest ventrally, so any small contribution of the deceleration signal to the peak changes would actually dilute the spatial gradient we report (Fig 2j,l). Finally, there were no significant correlations between the magnitude of the ITI movement response (min or max) and the magnitude of the changes for different ACh components during the task (across fibers with significant movement and task changes for each component, Fig. S7l,m, Table 1). Taken together with the analyses included in the initial version of the manuscript (Fig S7h-k), we therefore conclude that while locomotor decelerations can be associated with changes in ACh release at some striatal locations, these changes are unlikely to significantly effect or account for any of our main conclusions.

Minor:

When describing the behavioral indices of learning (Results paragraph ~lines 115-124), also mention & refer to Extended Data Fig 3 for learning to the tone cue too.

Author response (3.12). We have addressed this.

How many sessions on average were included as pre- vs post-learning? The mean (+/- SEM) for the group perhaps could be stated in passing in the Results text. How many intermediate sessions were excluded, where the behavioral lick index for one cue (but not both) was significantly different from the first session? Were such sessions primarily due to different

learning rates between cue modalities for a given animal, or differences in session 1 lick index values?

Author response (3.13). We have added the average sessions included for each phase and how many sessions were excluded to the text (lines 765-769). The differences between the learning rates for different cue modalities seem to be due to learning rate differences rather than initial lick index differences.

Double-check the wording/grammar in the sentence starting in line 98 (“Only a small fraction of fibers showed a sparse statistically significant changes...”) – may be just a singular/plural agreement issue, but something reads slightly off here. Also missing a closing parenthesis?

Author response (3.14). This has been corrected.

Add ‘Flex’ in the virus labels for the TeLC & iGluSnFR AAVs in the Fig 6a & 8a schematics? And consistent nomenclature for the tetanus toxin abbreviation: currently TeCl in Fig 6a, TelC most often throughout the text, and TeLC elsewhere.

Author response (3.15). We have made these corrections.

The sites targeted for these TelC, DA and glutamate experiments all appear fairly medial. Would it be more precise to label this as ‘aDMS’ rather than ‘aDS’ throughout?

Author response (3.16). It is correct that the injections were slightly medially shifted in the aDS, and we have edited the text to explicitly highlight this (lines 358-360). Sites were chosen to match the distribution of the late peaks observed for the light cue during extinction.

The blue colors used for the ‘Early peak only’ and ‘Dip and late peak’ locations are a bit difficult to differentiate (Figs 1h, 2h, 3c, 4e, perhaps more so for the text than for the dots themselves).

Author response (3.17). We have addressed this.

The legend for Fig 5e could help better clarify what the left vs right panel columns are (short vs long latency peaks?). Are the dip responses mapped in a figure somewhere for these unrewarded lick bouts? Post-learning responses (aside from the post-vs-pre differences in 5f)?

Author response (3.18). The purpose of this figure is to contrast the distribution of the early and late peaks in the pre-learning phase. We separated the plots for the early and late peaks to improve clarity. We agree that the labeling was not clear and have added labels for each column of the figure (Fig. S6b). The dip changes are shown in Fig. S6c.

Methods line 738: “DA<-4mm”, DA should be DV

Author response (3.19). This has been corrected.

Reviewer #4 (Remarks to the Author):

This manuscript explores the role of acetylcholine (ACh) in the striatum in behavioral flexibility during the learning and extinction of Pavlovian associations. The authors investigate ACh release and its multi-phasic response to both salient and reward-related cues, as well as rewards. Using an impressive novel technique that allows to record from up to 99 micro-fibers simultaneously, the authors sample ACh release across the whole extent of the striatum. The results show region-specific changes during different phases of learning and reward consumption and during extinction. ACh release related to reward delivery was found to be widespread but more restricted during the presentation of cues that signal reward. Notably, ACh release in the anterior dorsal striatum (aDS) shows a late peak during extinction, suggestive of a negative reward prediction error triggered by a decrease in dopamine-mediated inhibition. This study thus reveals significant differences in the dynamics of ACh across striatal regions and contributes to the mechanistic understanding of the role of ACh in behavioral flexibility. The quality of the data is exceptional, the experimental approach is rigorous, and the significance is high. However, the manuscript can be improved by organizing the results differently and clearly summarizing the main findings for each region and condition, as explained below.

Specific comments:

The manuscript is very rich in data, covering a large amount of event related responses in a variety of conditions, making the text difficult to follow. Several variables are presented, which include phases of training, behavioral events, types of cues and striatal regions. The authors use stereotaxic maps to represent changes in ACh across striatal regions but it is difficult to extract the significance of each plot. Simplified figures providing an average of the phase-specific changes in ACh release corresponding to the many different conditions described above would help the reader to get a comprehensive overview of the results and draw their own conclusions and make comparisons.

Related to the point above, the striatal regions sampled for each condition, illustrated by the fiber location, are not consistent across figures. The maps used to illustrate the changes in ACh signaling across striatal regions are repetitive and offer little helpful information. The description in the text is sparse and the patterns of topographical features scarcely described, making the text loosely related to the maps. Using the maps as supplementary figures and instead illustrate the changes in ACh release across a limited number of striatal regions (e.g., aDS, ventral striatum, tail of the striatum, etc.) would make the data clearer and the significance more evident.

Author response (4.1). We agree with the reviewer and have significantly re-organized and changed the figures. At the reviewer's suggestion, we now provide quantification of the direction and frequency of the phase specific changes for each signal component (early peak, dip, late peak), learning phase (initial learning, extinction), and task event across all fibers and individual mice (e.g. Fig 2g,i). This shows the overall trends in the signal changes independently of the spatial topography and that these changes are robust across fibers in individual mice. We have

also added new figure panels and analyses to more clearly display and quantify the topography of ACh dynamics across phases. As suggested, the maps showing changes in individual fibers have been moved to the supplement. The power of our large-scale array approach is that we are able to define the topography of signal changes without relying on arbitrary anatomical subregion boundaries. We have taken two new approaches to addressing this. First, to better capture and visualize the distribution of signal changes during learning and extinction, we replaced the maps in the main figures showing changes for each individual fiber with maps showing a smoothed weighted mean of changes collapsed across the axial and sagittal planes (e.g. Fig. 2j,k). From these maps, we defined regions over which the strongest concentration of changes occurred for each signal component (white contour lines). Second, to explicitly quantify and compare the variation of signal changes across all three striatal axes, we used a generalized linear model approach (see Methods, e.g. Fig 2l,m). This provided explicit statistical support for our claims that changes in different signal components occurred across distinct 3-dimensional spatial gradients during learning and extinction.

Regarding the calculation of dF/F (lines 87-89 and methods), by using such a low percentile (the 8th percentile) as the baseline, the authors are making the signal detection highly sensitive to small variations in the fluorescence signal. This means that even minor increases from this low baseline can be interpreted as significant changes, which might be just noise or minor fluctuations. This increased sensitivity raises the risk of misinterpreting noise or artifacts as genuine signals, especially if the baseline is unstable or exhibits significant fluctuations. While this approach may be justified if the goal is to detect subtle changes in fluorescence, it introduces the concern that these subtle changes could be mistaken for motion artifacts. The authors are addressing motion artifacts using a null-sensor from an external control group rather than an internal control. Given this setup, how can the authors be confident that the motion artifacts in the external control group are sufficiently similar to those in the experimental group? This is crucial for ensuring that subtle changes detected using the 8th percentile baseline are truly reflective of biological signals rather than artifacts.

Author response (4.2). To determine whether significant increases or decreases in signal were present around task events, we compared DF/F changes to the distribution of 'spontaneous' signals occurring outside of the task events (pre-cue ITIs). The 8th percentile values were only used to obtain the initial DF/F values, and this normalization was applied across all timepoints, inside and outside the task. Therefore, the exact percentile value used for the DF/F calculation does not impact the detection of statistically significant changes around task events relative to the non-task ITI periods. Moreover, the majority of our main findings were regarding changes between learning phases (e.g. post-learning vs extinction). These comparisons likewise are independent from the baseline percentile used for the DF/F calculation.

We primarily used a separate cohort of mice expressing the null sensor because, as mentioned in the manuscript, isosbestic illumination likely did not detect some of the slow artifactual changes. We have added additional quantification to the manuscript showing that the region specific changes we report were consistent across multiple fibers in individual mice (e.g. Fig. 2g

right panel). However, significant differences across phases for nearly all of the conditions were not present in any of the mice in the null cohort (e.g. Fig. S2h, top right panel).

Data for reversal learning is shown in Fig. 4 (panels a, b and c), but there is no description of the results in the text and no further mention in the manuscript.

Author response (4.3). The data for the reversal was included in all of the signal quantification for the extinction phase. The reversal was performed in order to compare signals for the two cue modalities in the same mice and cues were counterbalanced across groups to account for differences between the initial extinction and reversal phases. In general, the main effects (larger aDS late peak, smaller ventral early peak for the 20% cue) were similar for a given cue modality regardless of initial extinction or reversal phase (see Author Response 3.3). This does not rule out the possibility that more subtle signal differences may exist between these phases.

The correlation between the dopamine dip and the ACh late peak is in agreement with current literature, but this correlation seems to occur only in aDS. However, most dopamine neurons have been shown to decrease their firing during negative RPE thus releasing cholinergic interneurons from inhibition across most striatal regions. Why wouldn't CINs in other striatal regions show an increase in the amplitude of the late peak, as they do in aDS?

Author response (4.4). The reviewer raises a very interesting point. There are a few possible explanations for why increases in the ACh late peak during extinction are almost exclusively observed in the aDS. First, while DA dips encoding negative reward prediction errors have been observed in many DA cell body recordings (mainly to unexpectedly omitted rewards), the expression of the dip has been shown to be variable across neurons and in DA release across striatum regions (e.g. Dabney et al 2020, Tsutsui-Kimura et al 2020). Therefore, it is possible that light cue evoked DA dips are larger in aDS relative to other regions. Another possibility is that DA dips are widespread, but CINs in aDS are more sensitive to the dips than other regions (due to receptor expression, basal DA levels, etc). Finally, our iGluSNFr measurements indicate that the presence of a late peak in glutamate release in the aDS, which may be unmasked by the emergence of DA dips, meaning that the aDS elevations may depend on the conjunction of large DA dips and appropriately timed, likely region specific, glutamatergic input. To test these possibilities, future experiments will be necessary to resolve the spatial distribution of CIN glutamate input and DA dips across the striatum. We have added additional text to the discussion on this point (lines 516-531).

Lines 193-195: The authors state that “these correlations could not account for the patterns of cue and reward evoked release that we observed”, by showing that “the magnitude of the cue-evoked dF/F does not vary with deceleration magnitude” (Ext Data Fig 4). This does not rule out that Ach release might be sensitive to, or signaling locomotion, e.g. deceleration. It only shows that it does not distinguish between slow and fast deceleration. The authors should provide data showing Ach release at locomotor events (e.g., deceleration) during the ITI or spontaneous movements are not coinciding with cue onset.

Author response (4.5). We agree with the reviewer's point, which was also raised by Reviewer 3. We have addressed this with additional analysis of ACh release to spontaneous decelerations in the ITI period (Fig. S7, see Author Response 3.11). In summary, we found that while locomotor decelerations can be associated with changes in ACh release at some striatal locations, movement related signals are unlikely to significantly affect or explain any of our main conclusions (lines 309-349).

Lines 690: Was the statistical analysis conducted by adding all fibers corresponding to a region out of the 295 fibers that were analyzed, or were signals from fibers corresponding to a region for each animal first averaged and then averaged across animals? Please provide a table that indicates the number of fibers for each animal for each region.

Author response (4.6). In the submitted version of the paper, we did not conduct analyses on individual striatum regions. We have added new quantification of spatial in signal changes to the manuscript (See Response 4.1).

Line 312: "...these results indicate that DA signals in the aDS exhibit emerging dips to cues after partial extinction and spontaneous ITI licks prior to learning which precede ACh elevations in the same striatal region". Since these measurements come from different cohorts and related to the median 27 trials prior to lick change, this temporal correlation needs to be expressed more cautiously or shown within the same animal.

Author response (4.7). We agree with the reviewer that the wording of this phrase inaccurately implies that the DA dips precede ACh over trials during extinction learning. Given variable learning rates across mice, this relationship cannot be conclusively shown without simultaneous DA and ACh measurements in the same mouse. The sentence was meant to refer to the average latency of the DA dip trough relative to the ACh peak, which were consistent across mice and sessions (Fig 7e). We have edited the text to resolve the ambiguous wording (lines 392-393, 396-397, 506-508).

Lines 70-71 and 449: The text describes that the 55-99 optical micro-fibers (50um diameter) are distributed uniformly, collecting about 100um axially and 25um radially from each tip without overlap. The authors refer to Figure 1a,c, which does not really show how this volume is calculated and the methods solely provides references. Since this is crucial to the experiment, please provide a brief explanation how the volume was calculated.

Author response (4.8). The estimate of the approximate collection region of our fibers was described in detail in our recent paper on the multi-fiber array method (Vu et al, *Neuron* 2024, see Supplementary Fig 2a). Effective fluorescence collection volumes from optical fibers in photometry ('photometry efficiency') are a product of excitation and collection efficiency. These efficiency parameters, as a function of distance from the fiber tip, have been estimated analytically and measured empirically in non-scattering media and scattering brain tissue for different diameter fibers (50-200 micron) in several studies (Engelbrecht, *Opt Express* 2009; Tai, *J. Biomed Opt* 2007; Pisanello, *Front Neurosci.* 2019). Pisanello et al (*Front. Neurosci* 2019)

used empirical measurements in brain slices to calculate the photometry efficiency for 200 micron diameter fibers with NAs of 0.39 and 0.50 (see Figure 6 of Pisanello et al). Photometry efficiency drops to <10% beyond approximately 250 microns axially and 150 microns radially from the fiber tip of the 200 micron, 0.50NA fibers. They did not measure the photometry efficiency for 50 micron diameter fibers, but they did measure the collection efficiency in a quasi-transparent medium (Pisanello et al Fig 4). Collection volumes of the 50 micron fibers (NA 0.22) were approximately an order of magnitude lower than those for the 200 micron diameter fibers with >90% of collected photons within ~100 microns from the tip axially and ~50 microns radially. Since these calculations were only for collection efficiency (not incorporating excitation efficiency) and were not conducted in brain tissue, these values are an underestimate of the true photometry efficiency. Moreover, the true 'collection volume' for in-vivo recordings will be even lower because it is also a product of the signal to noise of the imaging system and excitation/emission properties of the fluorescent sensor. The minimum separation of fiber tips in our array is 220 microns in x/y and 250 microns in z, meaning that, based on the measurements and calculations above, there will be no (or very minimal) 'cross talk' across fibers. We have added included text in the results section to briefly clarify these points (lines 114-118).

Line 166: The authors state that "Changes to the initial and late peaks [at reward delivery] were more variable and less widespread" but write in line 173-175: "changes in the reward response with learning were slightly more widespread than for the cues, indicating that learning related changes to cues and rewards are partially decoupled and vary across regions". This sounds contradictory. Please clarify.

Author response (4.9). We agree that this wording is confusing. We meant to say 'learning related changes to *dips* at cues and rewards...' Dip changes were slightly more widespread to rewards relative to cues but changes to peaks were consistently expressed only for the cues. This text has been removed in the current version of the manuscript.

Line 187 does not refer to figure 2.

Author response (4.10). We have corrected this.

Line 211: "Elevations in late peaks following partial extinction were present for at least one fiber in 7 out of 8 mice for the light and 5 out of 8 mice for the tone (26/53, 49% and 17/53, 32%...)". How relevant is the signal measured from 1 fiber out of a minimum of 55 fibers and how do the authors get to the total numbers of 26 and 17? Please clarify.

Author response (4.11). This quantification has been removed from the revised version of the manuscript. We now include quantification of fiber numbers with significant changes from each mouse, illustrating that specific changes for each signal component are robust across fibers and mice (e.g. Fig 2g,i). As described in Author Response 4.1, we have incorporated a GLM approach to quantify gradients in signaling changes across axes (e.g. Fig. 2l,m). The model includes a mouse identity term to account for effects of variability across individual mice.

Extended data Fig 3h: The legend states that colors indicate significance but panel says something else.

Author response (4.12). This panel has been removed in the current version of the manuscript.

Line 376: “we found that persistent inhibition of ACh release from aDS cCINs prevented the down-regulations of licking during partial extinction (figure 6)”. How can the authors here infer causality?

Author response (4.13). We do not imply with this statement that the ACh elevations alone caused the down-regulation of licking, only that intact aDS ACh release is necessary. We have re-worded our conclusions from the TelC experiment in the revised manuscript (lines 374-376).

The map in Fig. 2h suggest a pattern where dip only neurons are spatially clustered in the pre-learning phase, but this is not mentioned in the manuscript. If the purpose of presenting maps is to illustrate distribution patterns, quantitative analysis should be used.

Author response (4.14). As stated above, we now provide quantification of the key spatial distribution patterns including breakdowns of the number of fibers and mice showing significant changes within each highlighted region. We include the ‘component presence’ maps to show the signal properties at different sites independently for each phase, but the primary claims we make regarding spatial distributions are related to the signal changes across learning and extinction phases. We agree that there appears to be ‘dips only’ at some locations in the dorsal striatum to the light cue prior to learning. These dips are relatively sparse, however, so we hesitate to make any claims regarding spatial clustering.

Fig. 6: Why are TelC animals different to controls in the percent time liking during the pre-learning phase. How is this impacting the results during post-learning and extinction?

Author response (4.15). The observed difference in licking between TelC and control animals during the pre-learning phase may indicate an impact of ACh silencing on non-contingent spout licking. Mice may be less capable, in general, of down-regulating unrewarded licking behavior. Importantly, this does not affect our main conclusions regarding intact initial associative learning and impaired extinction (Fig 6d). TelC mice still upregulated cue-associated licking with learning, despite having slightly higher cue-associated licking in the pre-learning phase. However, they were impaired in down-regulating cue-associated licking during extinction, relative to the post-learning phase (Fig 6c,f).

Reviewer #5 (Remarks to the Author):

Reviewer #6 (Remarks to the Author):
